# Lipocalin 2 regulates mitochondrial phospholipidome remodeling, dynamics, and function in brown adipose tissue in male mice

Hongming Su [1], Hong Guo [1], Xiaoxue Qiu [1], Te-Yueh Lin [1], Chao Qin[2], Gail Celio[3], Peter Yong [4], Mark Senders [3], Xianlin Han [2], David A. Bernlohr [4] & Xiaoli Chen [1] ✉

Mitochondrial function is vital for energy metabolism in thermogenic adipocytes. Impaired mitochondrial bioenergetics in brown adipocytes are linked to disrupted thermogenesis and energy balance in obesity and aging. Phospholipid cardiolipin (CL) and phosphatidic acid (PA) jointly regulate mitochondrial membrane architecture and dynamics, with mitochondria-associated endoplasmic reticulum membranes (MAMs) serving as the platform for phospholipid biosynthesis and metabolism. However, little is known about the regulators of MAM phospholipid metabolism and their connection to mitochondrial function. We discover that LCN2 is a PA binding protein recruited to the MAM during inflammation and metabolic stimulation. *Lcn2* deficiency disrupts mitochondrial fusion-fission balance and alters the acyl-chain composition of mitochondrial phospholipids in brown adipose tissue (BAT) of male mice. *Lcn2* KO male mice exhibit an increase in the levels of CLs containing long-chain polyunsaturated fatty acids (LC-PUFA), a decrease in CLs containing monounsaturated fatty acids, resulting in mitochondrial dysfunction. This dysfunction triggers compensatory activation of peroxisomal function and the biosynthesis of LC-PUFA-containing plasmalogens in BAT. Additionally, *Lcn2* deficiency alters PA production, correlating with changes in PA-regulated phospholipid-metabolizing enzymes and the mTOR signaling pathway. In conclusion, LCN2 plays a critical role in the acyl-chain remodeling of phospholipids and mitochondrial bioenergetics by regulating PA production and its function in activating signaling pathways.

Adipose tissue plays a central role in metabolic homeostasis, inflammation, and insulin resistance. Mitochondrial bioenergetics of brown and beige adipocytes is an essential component of energy balance regulation. Mitochondrial dysfunction is linked to homeostatic disruption of adipose tissue metabolism and function in obesity as well as type 2 diabetes, and additionally contributes to age-related decline in thermogenesis of adipose tissue. The function of mitochondria requires the integrity of the mitochondrial membrane, which relies on a coordinated supply of proteins and phospholipids, the primary lipids in mitochondrial membranes. Of the total mitochondrial

phospholipids, phosphatidylcholine (PC) and phosphatidylethanolamine (PE) account for 40-45% and 25-30%, respectively. Cardiolipin (CL) and phosphatidylinositol (PI) comprise 10-15%, whereas phosphatidic acid (PA) and phosphatidylserine (PS) contribute 3-5% of total phospholipids in mitochondria[1–3]. While the majority of mitochondrial phospholipids are imported from the endoplasmic reticulum (ER), including PC, PI, and PS, some lipids are synthesized in the mitochondria such as CL, phosphatidylglycerol (PG), and PE[4,5].

Phosphatidic acid participates in the biosynthesis of phospholipids through a CTP-dependent activation catalyzed by CDP-diacylglycerol synthase. This enzyme forms CDP-diacylglycerol, a direct precursor for the biosynthesis of CL, PI, and PG. Additionally, PA is a key intermediate in the biosynthetic pathway known as the Kennedy pathway for the biosynthesis of triacylglycerol (TAG) and other glycerophospholipids PC and PE. CL is a class of mitochondria-specific phospholipids with a unique structure of four acyl chains and two phosphatidyl moieties linked to glycerol. Recent studies have suggested the essential role of CL and its precursor PA in the control of mitochondrial dynamic processes. PA and CL are involved in various membrane-related processes including the assembly and import of mitochondrial proteins as well as the formation of inter-organelle membrane contact sites[6,7]. Increasing evidence has revealed that CL and PA are directly involved in mitochondrial fusion and fission via regulating the assembly of mitochondrial fusion protein OPA1 and fission protein DRP1[6,8].

The ER and mitochondrial contact sites, also called mitochondria-associated ER membranes (MAMs) contain several lipid-biosynthesizing enzymes and play an important role in phospholipid biosynthesis and transport[9–12]. MFN2 and DRP1 are found to be enriched in MAMs and proposed to tether the ER and mitochondria[13,14]. Additionally, there exist lipid-binding proteins that directly promote inter-organelle lipid transfer[15]. However, such lipid-binding proteins have not yet been identified in mammalian cells. Impaired MAM integrity and function has been associated with the dysregulation of phospholipid metabolism leading to alterations in mitochondrial protein imports, mitochondrial architecture and dynamics, and mitophagy[7]. Most importantly, dysfunction of MAMs has been linked to multiple pathologies such as cellular aging[16] and obesity[17], activation of inflammasome[18], and disruption of systemic lipids and glucose homeostasis[19,20]. However, little is known about the regulation of phospholipid metabolism at the MAMs, the impact of phospholipid dysregulation on mitochondrial architecture/function, and what factors are involved in the regulation of these processes.

In this study, we discovered that LCN2 is a PA-binding protein and is recruited to the MAM during inflammation and metabolic stimulation. *Lcn2* deficiency impairs mitochondrial fusion leading to the fragmentation and swelling of mitochondria. *Lcn2* KO mice display increased long-chain polyunsaturated fatty acids (LC-PUFAs) acyl-chain remodeling of CLs, increased PUFA CL to MUFA CL ratio and oxidized CLs leading to mitochondrial damage and dysfunction in BAT. Consequently, peroxisome function is compensatorily activated during metabolic stimulation, as evidenced by increased LC-PUFAs and plasmalogen biosynthesis. We also demonstrated that *Lcn2* deficiency alters PA production and function as a recursive regulator of PA-producing enzymes, deregulates PA signaling pathway activity, activates mitochondrial dysfunction-associated inflammation, and increases oxidative stress in BAT.

## Results

### LCN2 is a phosphatidic acid binding protein and recruited to the MAM during inflammation and metabolic stimulation

We have previously demonstrated that LCN2 plays a role in thermogenic and mitochondrial function in brown and beige adipocytes[21–24], and identified LCN2 in crude mitochondrial fraction from adipocytes treated with lipopolysaccharide (LPS). To locate LCN2 in organelle membranes precisely, we separated pure mitochondria and MAMs from crude mitochondrial fraction using a Percoll-based subcellular fractionation method as previously reported[25]. Interestingly, LCN2 was detected abundantly in the MAM fraction in the pro-inflammatory state (Fig. 1a, b). LCN2 levels at MAMs are comparable to FACL4, a MAM-enriched protein, but much higher than other known MAM proteins such as DRP1 and MFN2 in inguinal adipocytes with 6 h treatment of LPS (Fig. 1a). Similar results were obtained in the MAMs isolated from brown adipose tissue (BAT) of mice after LPS injection for 6 h (Fig. 1b). Moreover, we showed increased MAM and mitochondrial localization of LCN2 in BAT of mice with 3 h treatment of CL316, 243, β3-adrenergic receptor agonist (Fig. 1c). To provide additional evidence supporting the MAM localization of LCN2, we performed double immunostaining to examine the co-localization of LCN2 with ER and mitochondria under confocal microscope. 3T3-L1 preadipocytes and adipocytes at day 5 of differentiation were treated with LPS for 4 h, followed by double immunostaining with antibodies against LCN2 and calnexin, an ER marker or TOM 20, a mitochondrial marker. We observed a strong colocalization of LCN2 with either ER (Fig. 1d) or mitochondria (Fig. 1e) with a high correlation coefficient.

One of the most important functions of MAMs is known to serve as the platforms for phospholipid exchange between ER and mitochondria[11]. This led us to investigate the potential binding of LCN2 to membrane lipids. First, we performed the lipid-binding assay on commercially available membrane lipid strips spotted with indicated lipids (Fig. 1f). We found that LCN2 strongly and specifically binds to phosphatidic acid (PA) (Fig. 1f). To further validate the specificity of LCN2 binding to PA, we conducted a concentration-dependent binding of LCN2 to PA. We spotted various doses of PA and PC on the nitrocellulose membrane as indicated in Fig. 1g and conducted the binding assay with LCN2. Similarly, we observed that LCN2 had a strong and specific binding to PA in a concentration-dependent manner (Fig. 1g). Moreover, we used a different approach to confirm PA-LCN2 binding and performed protein pull-down assay with PA-coated beads. Western blotting of LCN2 on proteins pulled down from PA beads clearly showed that LCN2 binds to PA in a concentration-dependent fashion (Fig. 1h). Together, these results suggest a potential role of LCN2 as a PA-binding protein.

### *Lcn2* deficiency impairs mitochondrial dynamics leading to mitochondrial fragmentation and swelling

MAMs play a critical role in the regulation of mitochondrial membrane lipid homeostasis, mitochondrial fusion/fission, and mitochondrial function[16,18,26]. MAM phospholipid (CL) metabolism determines the formation of mitochondrial cristae and mitochondrial dynamics and function[27–29]. To determine the impact of *Lcn2* deficiency on mitochondrial dynamics, we used MitoTracker probe to examine the morphological dynamics of mitochondria in *Lcn2* KO adipocytes under the fasting-refeeding cycle. Stromal-vascular (SV) cells were isolated from brown adipose tissue (BAT) of WT and *Lcn2* KO mice and induced to differentiate into brown adipocytes. Differentiated brown adipocytes were treated with EBSS or EBSS followed by high glucose combined with palmitate acid (HG + PA) to induce fusion and fission. Under the confocal microscope, we observed tubular mitochondrial shape (resulting from fusion) in the fasting state, but small punctate structures (resulting from fission) in the refeeding state in WT brown adipocytes (Fig. 2a). However, in *Lcn2* KO brown adipocytes no significant tubular structures were observed during the fasting state; mitochondria constantly showed small punctate structures in both fasted and refed states (Fig. 2a). These results strongly suggest that *Lcn2* deficiency impairs mitochondrial fusion leading to mitochondrial fragmentation. As shown in Fig. 2b, refeeding with HG + PA increased DRP1 phosphorylation at Ser616 in WT brown adipocytes, but at a much lower level in *Lcn2* KO cells. *Lcn2* deficiency had no significant effect on

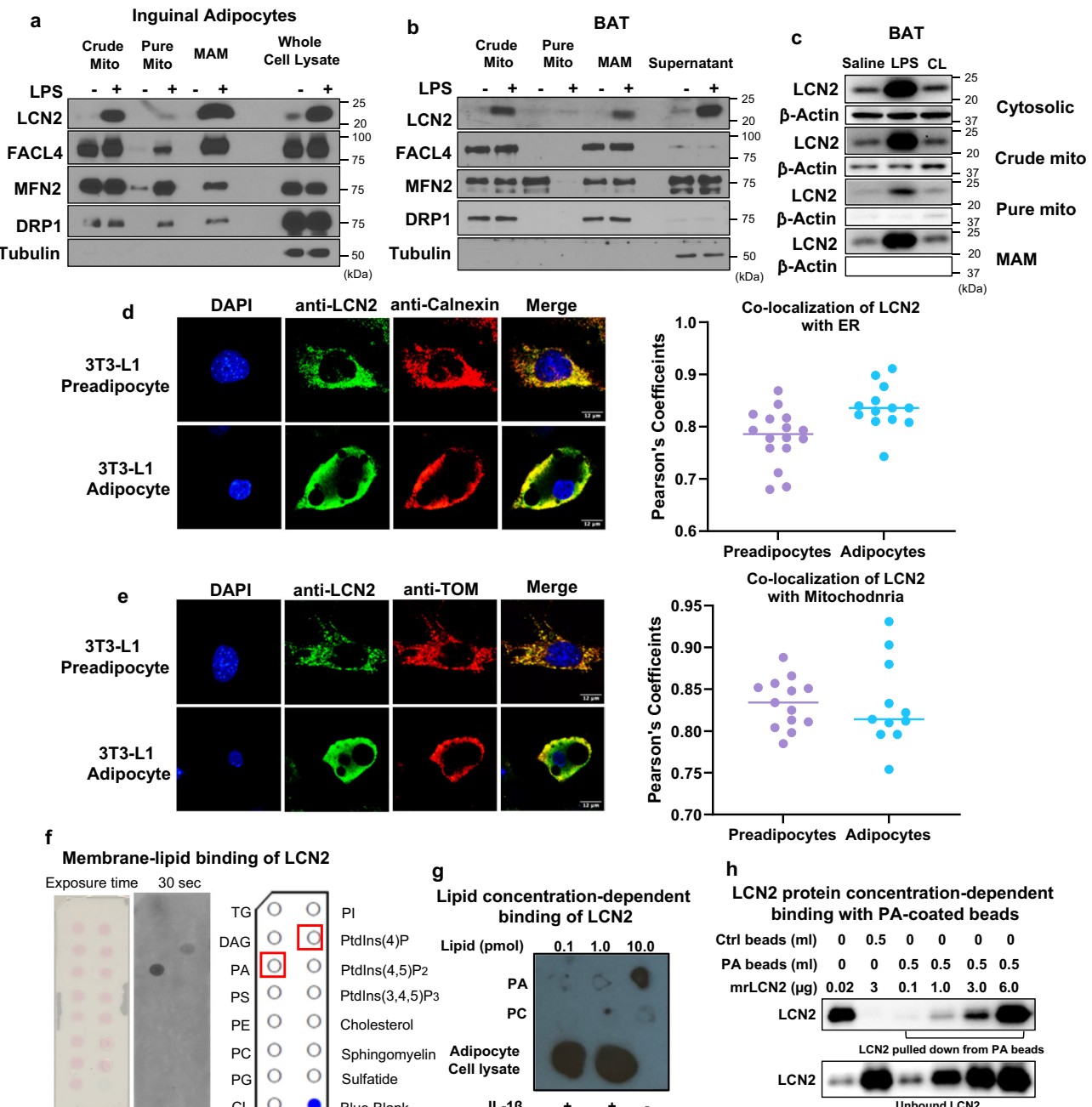

**Fig. 1 | Identification of LCN2 MAM localization and PA binding capability.** LCN2 is recruited to the MAM in inguinal adipocytes (**a**) in response to 6 h LPS (1 µg/mL) treatment and in BAT (**b**) in response to 6 h LPS stimulation along with known MAM proteins including FACL4, MFN2 and DRP1. The experiments in a and b were repeated 3 times independently. LCN2 protein levels in cytosolic, crude mitochondria, pure mitochondria, and MAM fractions in BAT of mice with saline, 6 h LPS (0.3 mg/kg body weight), or 6 h CL316, 243 (0.5 mg/kg body weight) treatment (**c**). The experiment in c was repeated twice independently. Co-localization of LCN2 with ER marker Calnexin (**d**) and mitochondrial maker TOM 20 (**e**) in 3T3-L1 pre-adipocytes and adipocytes. The correlation analysis was performed using Pearson's coefficient. For the co-localization of LCN2 with ER, *n* = 16 for preadipocytes and

*n* = 13 for adipocytes (**d**). For the co-localization of LCN2 with mitochondria, *n* = 13 for preadipoctes and *n* = 11 for adipocytes (**d**). Results are presented as mean ± SEM. Binding assay of membrane lipids with mouse LCN2 recombinant protein (**f**). Concentration-dependent binding of LCN2 to phospholipids (**g**). Various concentrations of PA and PC were spotted on the membrane and subjected to LCN2 binding assay. Cell lysates from 3T3-L1 adipocytes with or without IL-1β (1 ng/mL) treatment were spotted as a positive control. Concentration-dependent LCN2-PA binding with PA-coated beads (**h**). Beads without PA-coated served as a control. The experiment in h was repeated twice independently. Source data are provided as a Source data file. Mito: mitochondria; CL: CL316, 243.

OPA1 and MFN2 levels under the fasted and fed state (Fig. 2b). Further, norepinephrine (NE) or CL316, 243 (CL) treatment for 1 h induced DRP1 phosphorylation at a lower level in *Lcn2* KO brown adipocytes compared to WT controls (Fig. 2c). Together, these results suggest that LCN2 is essential for the regulation of mitochondrial dynamics, particularly the fusion process.

Next, we performed transmission electron microscopy (TEM) to analyze the mitochondrial morphology in BAT from mice exposed to 23 °C or 4 °C for 5 h, as acute cold challenge is known to induce mitochondrial dynamics and biogenesis in BAT. As illustrated in Fig. 2d, there was no significant difference in mitochondrial size between WT and *Lcn2* KO brown adipocytes from

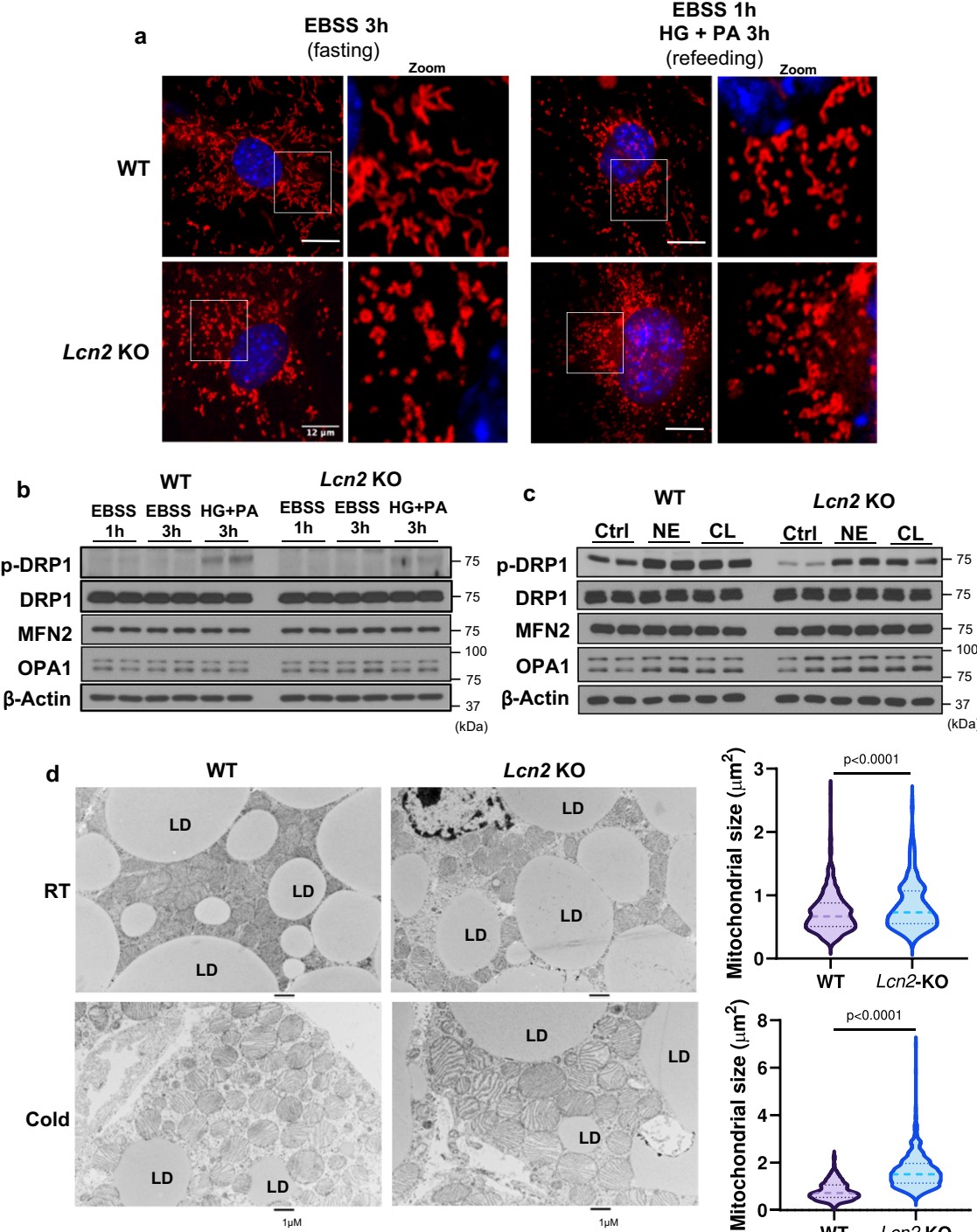

**Fig. 2 | Effect of *Lcn2* deficiency on mitochondrial dynamics.** Mitochondrial dynamics induced by fasting and refeeding cycle in differentiated brown adipocytes (**a**). The expression of proteins involved in mitochondrial fission and fusion in differentiated brown adipocytes during the fasting-refeeding cycle (**b**) and upon norepinephrine (NE) (1 μM) or CL316, 243 (CL) (1 μM) treatment for one hour (**c**). TEM analysis of mitochondrial morphology and quantification of mitochondrial size in brown adipose tissue of mice at room temperature (RT) or after 5 h of 4 °C cold exposure (**d**). The in vitro experiments in b and c were repeated independently 3 and 2 times, respectively. The average mitochondrial size was determined based on 402 and 392 mitochondria from WT and *Lcn2* KO mice at room temperature, and 496 and 510 mitochondria from WT and *Lcn2* KO mice after 5 hours of exposure to 4 °C cold temperature (*n* = 3). Results are presented as mean ± SEM. Statistical significance was analyzed by Student's *t*-test; all tests were two-sided. Source data are provided as a Source data file. PA: palmitate acid; HG: high glucose; LD: lipid droplet.

mice at the ambient temperature. However, after cold exposure, *Lcn2* KO brown adipocytes had increased mitochondrial size as well as larger lipid droplets compared to WT cells. Using an Image J program, we quantified the size of 496 mitochondria from WT adipocytes and 510 *Lcn2* KO mitochondria. The results showed

that *Lcn2* KO brown adipocytes had significantly larger mitochondria with an average size of approximately 1.6 μm² compared to those with the size of 0.8 μm² on average in WT cells (Fig. 2d). Along with the mitochondrial fusion and fission morphology from in vitro differentiated brown adipocytes, these results suggest that

*Lcn2* deficiency impairs mitochondrial dynamics, leading to mitochondrial swelling and fragmentation.

### *Lcn2* deficiency blunts the metabolic effect of CL316, 243 in BAT and Ing-WAT

CL316, 243 is known to induce mitochondrial biogenesis and function in BAT, the beiging of inguinal white adipose tissue (Ing-WAT), and thermogenesis[30–32]. To investigate the role of *Lcn2* in mitochondrial metabolism and metabolic homoestasis, WT and *Lcn2* KO mice were treated with either saline or CL316, 243 (0.5 mg/kg BW) via i.p. injection once a day for 14 days. While the body weight was not significantly changed in *Lcn2* KO mice, WT mice lost about one gram of body weight on average during 14 days of CL316, 243 treatment (Supplementary Fig. 1a). The weight of fat depots was not significantly changed by CL316, 243 treatment in WT mice, whereas CL316, 243 treatment increased both inguinal and epididymal fat mass in *Lcn2* KO mice (Supplementary Fig. 1b–d). After CL316, 243 treatment, *Lcn2* KO mice had significantly lower levels of serum insulin, but higher levels of blood glucose compared to WT mice (Supplementary Fig. 1e, f). Serum-free fatty acid levels had a decreasing trend in WT mice by CL316, 243, but no change in *Lcn2* KO mice (Supplementary Fig. 1g). The histological morphology of adipose tissues showed that CL316, 243 treatment reduced the size of lipid droplets in brown adipocytes from WT mice (Supplementary Fig. 1h). However, CL316, 243-treated *Lcn2* KO mice had increased lipid accumulation and showed slightly larger brown adipocytes compared with WT controls (Supplementary Fig. 1h). In Ing-WAT, CL316, 243 treatment for 14 days caused a significant change in adipocyte morphology, i.e. switching from unilocular to multilocular adipocytes in WT mice, indicating the beiging of Ing-WAT by CL316, 243 (Supplementary Fig. 1h). Intriguingly, the CL316, 243-induced beiging effect was blunted in *Lcn2* KO mice (Supplementary Fig. 1h). In epididymal adipose tissue (Supplementary Fig. 1h), CL316, 243 treatment caused a slight reduction in adipocyte size similarly in both WT and *Lcn2* KO mice (Supplementary Fig. 1h). Collectively, these results suggest that *Lcn2* is required for the metabolic effect of β3-AR signaling activation in BAT and Ing-WAT.

### *Lcn2* deficiency alters CL316, 243-induced reprograming of mitochondrial phospholipidome in BAT

Activation of thermogenesis with CL316, 243 has been shown to regulate phospholipid metabolism in BAT[33]. Therefore, we performed lipidomics analysis of crude mitochondria isolated from BAT of mice treated with saline or CL316, 243 for 14 days. We were able to quantify 246 lipid species including PC, PE, sphingomyelin (SM), CL, lyso cardiolipin (MLCL), PA, PS, PG, PI, various lysophospholipids, acylcarnitine (CAR), oxyl-cardiolipin (O-CL), triacylglycerol (TAG), and free fatty acids (FFA) (Supplementary Data 1). Principle component analysis (PCA) on concentrations of 246 mitochondrial lipids showed that WT and *Lcn2* KO clusters overlapped in the saline-treated condition, but well separated in the CL316, 243-treated condition (Supplementary Fig. 2a), suggesting that WT and *Lcn2* KO mice have a different response to CL316, 243 in mitochondrial phospholipid metabolism. This is confirmed by PCA on the fold change (calculated from the ratio of CL316, 243- to saline-treated concentrations of lipids) (Supplementary Fig. 2b). Furthermore, hierarchical clustering showed a differential pattern of CL316, 243-induced changes in mitochondrial phospholipidome (Supplementary Fig. 2c). Together, these results suggest that *Lcn2* deficiency alters CL316, 243-induced mitochondrial phospholipidome in BAT.

According to PCA on specific categories of lipids, the most significant difference between WT and *Lcn2* KO mice was found in CL316, 243-induced fold changes in CL and MLCL. We next investigated how *Lcn2* deficiency affects CL biosynthesis and metabolism. Although there was no genotypic difference in the sum of total 26 measurable CL species in either saline- or CL316, 243-treated condition (Fig. 3a), both

PCA and hierarchical clustering showed that CL316, 243 induced a distinct pattern of changes in the content of these CL species in *Lcn2* KO BAT (Fig. 3b, c). We then categorized the 18 CL species that were changed in *Lcn2* KO mice based on their change patterns. In the first category, five FA (16:1)-containing CL species (Fig. 3d, h) had decreased basal levels in *Lcn2* KO BAT; CL316, 243 reduced their levels in WT but unchanged in *Lcn2* KO BAT. The second category of FA (16:1)-containing CL species were significantly reduced by CL316, 243 in *Lcn2* KO but not in WT BAT (Fig. 3i–m). The third category of FA (C18 and C20-22)-containing CLs had significantly higher basal levels in *Lcn2* KO BAT, and CL316, 243 was able to increase the levels of these CL species in WT and to a lesser extent in *Lcn2* KO mice (Fig. 3n–u).

Since the production of mature CLs involves multiple steps including the biosynthesis of MLCL and the remodeling of CL, it is of importance to determine how *Lcn2* deficiency changes the levels of MLCL. Although the total levels of measurable MLCL species were not genotypically different (Supplementary Fig. 3a), both PCA and hierarchical clustering indicated that WT and *Lcn2* KO mice had a different response to CL316, 243 in MLCL content changes (Supplementary Fig. 3b, c). For instance, CL316, 243 significantly increased the total MLCL levels in WT mice but not in *Lcn2* KO mice (Supplementary Fig. 3a). Moreover, we demonstrated that the levels of nine MLCL species were differentially altered in response to CL316, 243 in WT and *Lcn2* KO mice (Supplementary Fig. 4d–l). Interestingly, these altered MLCLs mostly contain C18 and C20-22 FAs. No changes were observed with MLCLs-containing FA 16:1.

### *Lcn2* deficiency remodels the acyl composition of cardiolipins, PC, PE, and PS in crude mitochondria isolated from BAT

Since fatty acyl-chain composition of CLs plays a more important role in mitochondrial function than the total content of cardiolipins[34,35], we next determined the fatty acyl-chain composition of CLs. Measurable CL species were categorized into three groups, i.e. MUPA CLs (CLs containing saturated and monounsaturated fatty acids), C18:2n6 PUFA CLs (CLs containing C18:2 PUFA), and C20-22 PUFA CLs (CLs containing C20 or C22 PUFA). First, the PCA analysis of each CL category showed that *Lcn2* deficiency significantly altered the effect of CL316, 243 on the content of MUFA and C20-22 PUFA but not C18:2 PUFA CLs (Fig. 4a–c). The sum of measurable CL species in each category showed that CL316, 243 significantly reduced the sum of MUFA CLs in BAT of *Lcn2* KO but not WT mice (Fig. 4a). In spite of this, the sum of MUFA CLs had no significant phenotypic difference in the basal and a decreasing trend in the CL316, 243-treated condition (Fig. 4a). The sum of C18:2 PUFA CLs had no significant difference between WT and *Lcn2* KO BAT in either basal or CL316, 243-treated condition (Fig. 4b). Intriguingly, *Lcn2* KO mice had significantly higher basal levels of sum C20-22 PUFA CLs (including both C20-22n6 PUFA and C20-22n3 PUFA CLs) compared to WT mice (Fig. 4c). CL316, 243 increased C20-22-PUFA CLs in both WT and *Lcn2* KO mice (Fig. 4c).

Since newly synthesized CLs are mainly MUFA CLs containing 16:0 and 18:1 FAs and increased PUFA CLs is indicative of an increased CL remodeling, it would be of importance to know the relationship between MUFA CLs and PUFA CLs in terms of their composition changes. To this end, we determined the percentage of each category of CL species in the total measurable CLs. After CL316, 243 treatment, the percentage of MUFA-CLs was markedly reduced (Fig. 4d), whereas the percentage of C18:2n6 PUFA CLs was significantly increased in *Lcn2* KO mice compared to WT mice (Fig. 4e). CL316, 243 had no significant effect on the percentage of these two categories of CLs in WT mice (Fig. 4d, e). Intriguingly, *Lcn2* KO mice had significantly higher percentage of C20-22n6 PUFA CLs than WT mice in the basal condition; CL316, 243 increased the percentage of C20-22n6 PUFA CLs in both WT and *Lcn2* KO BAT (Fig. 4f). However, the percentage of C20-22n3 PUFA CLs was not different between genotypes (Fig. 4f). Overall, these changes lead to a decrease in MUFA-CLs and an increase in PUFA CLs,

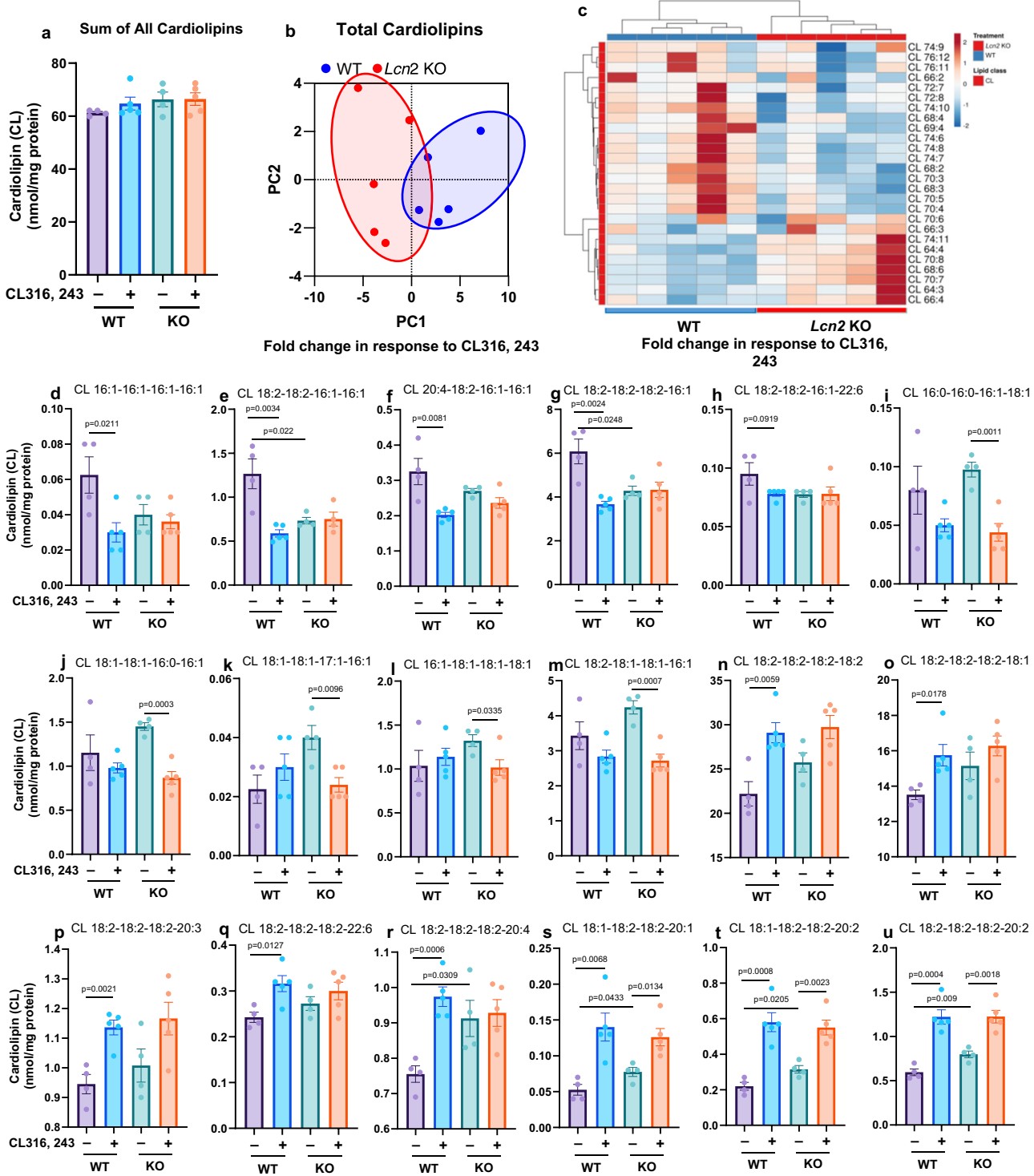

**Fig. 3 | Effect of *Lcn2* deficiency on CL316, 243-induced changes in cardiolipin content in BAT.** WT and *Lcn2* KO mice at 8–10 weeks of age were treated with either saline or CL316, 243 (0.5 mg/kg BW) via i.p. injection once a day for 14 days. Global analysis of total cardiolipins (**a**), PCA (**b**), and hierarchical clustering (**c**). CL316, 243-induced changes in the content of individual cardiolipins (**d-u**). Results are presented as mean ± SEM. Student's *t*-test was performed to test differences between two independent groups. All tests were two-sided. *n* = 4 (saline-treated WT and *Lcn2* KO mice), *n* = 5 (CL316, 243-treated WT and *Lcn2* KO mice). Source data are provided as a Source data file.

thereby an increased PUFA CL to MUFA CL ratio in *Lcn2* KO BAT (Fig. 4g). Further analysis of the percentage of individual CL species that were changed in *Lcn2* KO BAT showed that FA16:1-containing CLs were significantly decreased in *Lcn2* KO BAT in either basal or CL316, 243-treated condition (Fig. 4h). Specifically, FA 16:1-containing MUFA CLs had higher basal levels, but were more profoundly reduced by

CL316, 243 in *Lcn2* KO mice compared to WT controls. Nevertheless, FA 16:1-containing PUFA CLs had reduced basal levels but remained unchanged after CL316, 243 treatment. Because of significantly decreased by CL316, 243 in WT mice, FA 16:1-containing PUFA CLs had higher levels in *Lcn2* KO mice in the CL316, 243-treated condition. Strikingly, LC-PUFA-containing CLs were significantly increased in *Lcn2*

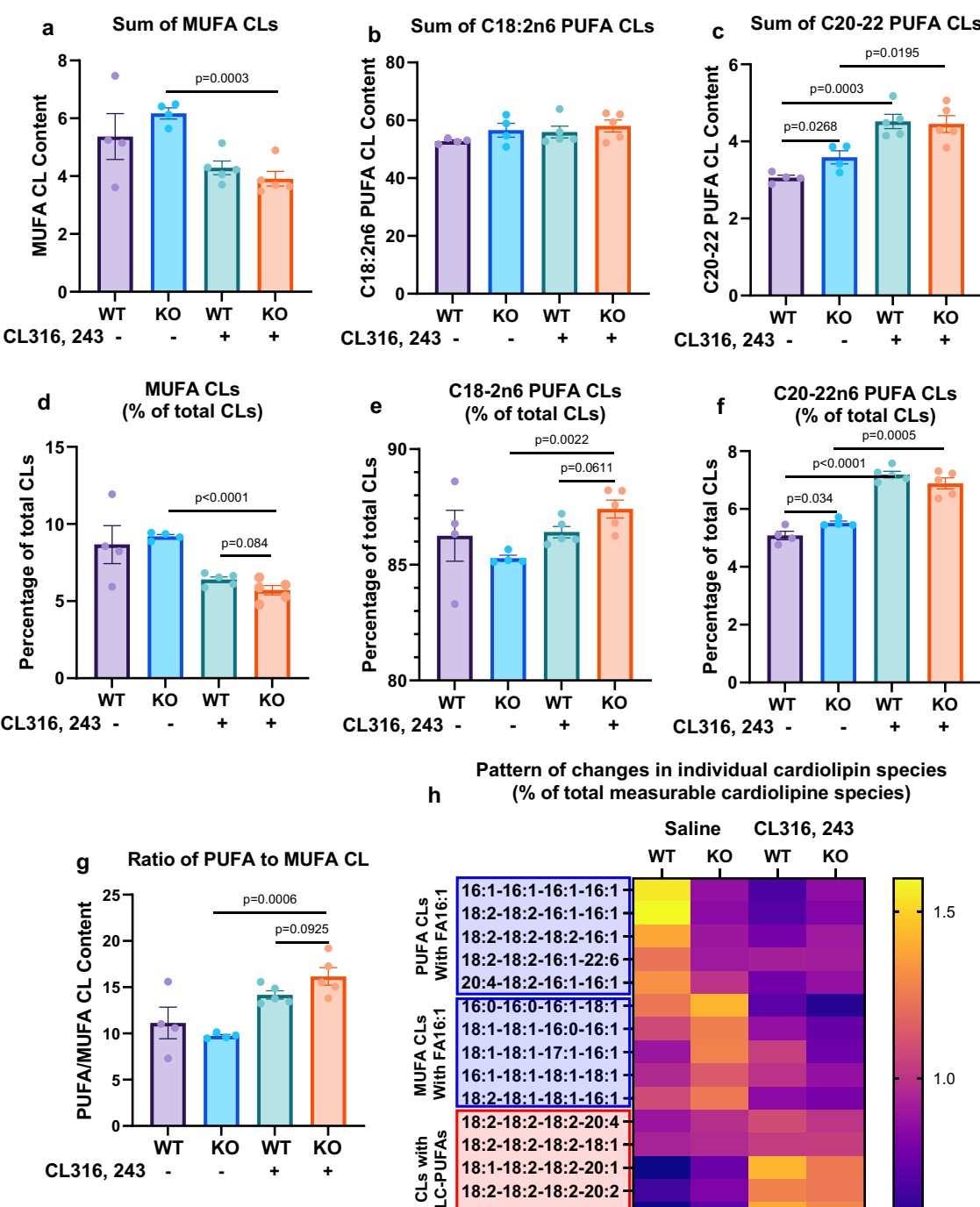

**Fig. 4 | Global analysis of three categories of measurable cardiolipin species in BAT.** (**a**) Sum of MUFA-containing CL species, C18:2n6 PUFA-containing CL species (**b**), C20-22 PUFA-containing CL species (**c**), Percentage of MUFA-containing cardiolipin (**d**), C18:2n6 PUFA-containing cardiolipin (**e**), and C20-22n6 PUFA-containing cardiolipin (**f**) in total measurable cardiolipin species in mitochondria isolated from BAT, and ratio of PUFA to MUFA-containing CL species (**g**) in BAT of male mice treated with or without CL316, 243 for 14 days. Pattern of changes in individual CL species presented as a percentage of total measurable CL species (**h**). Results are presented as mean ± SEM. Student's *t* test was performed to test differences between two independent groups. All tests were two-sided. *n* = 4 (saline-treated WT and *Lcn2* KO mice), *n* = 5 (CL316, 243-treated WT and *Lcn2* KO mice). Sum of MUFA-CLs was calculated from the addition of the abundance of 11 measurable MUFA CLs (16:1-16:1-16:1-16:1, 16:0-16:1-16:1-16:1, 16:1-16:1-16:1-18:1,

16:0-16:1-16:1-18:1, 16:0-16:0-16:1-18:1, 18:1-18:1-16:1-16:1, 18:1-18:1-16:0-16:1, 18:1-18:1-16:0-16:0, 18:1-18:1-17:1-16:1, 16:1-18:1-18:1-18:1, 16:1-18:1-18:1-18:0). Sum of C18:2n6 PUFA-CLs was calculated from the addition of the abundance of 7 measurable C18:2n6 PUFA-CLs (18:2-18:2-16:1-16:1, 18:2-18:2-18:2-16:1, 18:2-18:2-18:2-16:0, 18:2-18:1-18:1-16:1, 18:2-18:2-18:2-18:2, 18:1-18:2-18:2-20:1, and 18:2-18:2-18:2-18:1). Sum of C20-22 PUFA-CLs was calculated from the addition of the abundance of 6 measurable C20-22n6 PUFA-CLs (20:4-18:2-16:1-16:1, 18:2-18:2-18:2-20:4, 18:2-18:2-18:2-20:3, 18:2-18:2-18:2-20:2, 18:1-18:2-18:2-20:2, and 18:2-18:2-18:2-22:5) and 2 measurable C22:6 PUFA CLs (18:2-18:2-16:1-22:6 and 18:2-18:2-18:2-22:6). The percentage of MUPA-CLs, C18-2n6 PUFA CLs, and C20-22n6 PUFA CLs was calculated from the sum of each category of CLs/total measurable CLs. Source data are provided as a Source data file.

KO mice compared to WT mice (Fig. 4h). All the results above suggest that *Lcn2* deficiency affects more the remodeling than the de novo biosynthesis of CLs, as increased PUFA-containing CLs are reflective of the increased remodeling of CLs[36]. The reduction in MUFA CLs particularly CLs containing FA 16:1 in *Lcn2* KO mice is likely due to an increased remodeling of LC-PUFA CLs, leading to the impaired balance between PUFA and MUFA CLs, as reflected by the increased PUFA-CL to MUFA-CL ratio. Further, we conducted General Linear Models Analysis to determine the contribution of *Lcn2* deficiency to changes in the CL remodeling. We showed that there was significance in genotype and treatment interaction in the percentage of C20-22 LC-PUFA CLs (Supplementary Fig. 4a–c), supporting that *Lcn2* deficiency leads to the acyl-chain remodeling of cardiolipins with increased LC-PUFA incorporation in BAT.

Next, we examined the gene and protein expression of enzymes involved in CL biosynthesis and remodeling. While the gene expression levels of enzymes involved in CL biosynthesis and remodeling including *Pgs1*, *Ptpmt1*, *iPla2γ*, and *Taz* were not significantly different between genotypes and treatments (Supplementary Fig. 5a, b, e, f), the expression levels of CL biosynthesis enzymes *Crls1* and *Tamm41* were significantly upregulated by CL316, 243 in BAT of WT mice but not in *Lcn2* KO mice (Supplementary Fig. 5c, d). However, the protein level of CRLS1 was increased by CL316, 243 stimulation in both WT and *Lcn2* KO mice (Supplementary Fig. 5g–j). Tafazzin (TAZ), a CL remodeling enzyme that converts MLCL to (18:2)4CL was not genotypically different in the basal and CL316, 243-treated condition (Supplementary Fig. 5g–j). ALCAT1, another CL remodeling enzyme having a role in the pathological remodeling of DHA-containing CL[37] was upregulated by CL316, 243 in both WT and *Lcn2* KO mice to a similar extent (Supplementary Fig. 5g–j). These data further support that *Lcn2* deficiency affects more the CL remodeling process, which contributes to the alteration of CL acyl-chain composition through a TAZ- and ALCAT1-independnet mechanism.

PC and PE, the important intermediates in the CL remodeling process are synthesized from both Kennedy and PEMT (phosphatidylethanolamine *N*-methyltransferase) pathways. Additionally, Land's cycle is involved in the synthesis of C20:4-containing PC[38]. To understand how *Lcn2* deficiency affects the CL remodeling, we examined the acyl composition of PC, PE, and PS, as well as the enzymes involved in the above-mentioned pathways. The sum of measurable PC, PE, and PS was not different between genotypes. Interestingly, similar to changes in the FA composition of CL species, the basal levels of LC-PUFA-containing PCs (Fig. 5a–d) and FA(20:4)-containing PE (Fig. 5g) were significantly increased in *Lcn2* KO BAT compared to WT BAT, and CL316, 243 raised the levels of these lipid species in both WT and *Lcn2* KO BAT (Fig. 5a–d, g). The basal levels of FA (16:1)-containing PCs were decreased in *Lcn2* KO BAT (Fig. 5e, f). LC-PUFA-containing PS species generally had increased levels in the CL316, 243-treated condition (Fig. 5h–k). Moreover, the sum of LPCs was significantly decreased in *Lcn2* KO versus WT BAT with CL316, 243 treatment (Fig. 5l). Specifically, CL316, 243-induced LPC (20:4) and LPC (22:6) but not LPC (18:0) were significantly lower in *Lcn2* KO BAT compared to WT BAT (Fig. 5m–o). Looking at the mRNA expression of enzymes involved in PC and PE biosynthesis (Supplementary Fig. 6i), the expression of enzymes involved in MUFA-containing PCs that are known to be primarily via Kennedy pathway[38] was not genotypically different (Supplementary Fig. 6a–e). However, PEMT responsible for the synthesis of PUFA-containing PCs in the PEMT pathway as well as LPCATs such as *Lpcat4* in the Land's cycle for the remodeling of C20:4-containing PC[38,39] had significantly higher expression levels in the CL316, 243-treated condition (Supplementary Fig. 6f–j). Collectively, the data suggests that *Lcn2* deficiency increases the LC-PUFA remodeling of PC, PE, and PS.

## *Lcn2* deficiency increases oxidized cardiolipin, acyl carnitine, and mitochondrial damage-associated inflammation in BAT

The fatty acid desaturation of complex lipids has been associated with aging in humans and mice. The desaturation of fatty acid chain in phospholipids affects the solubility and fluidity of membranes as well as the lipid susceptibility to oxidative damage. Due to the presence of multiple double bonds, PUFAs are more susceptible to oxidative stress than MUFAs. The PUFA to MUFA ratio in membrane phospholipids is known to increase with age[40–44]. Previous studies also showed that PUFA CLs are increased, whereas 16:1 FA-containing CLs are markedly decreased in myocardium of obese and diabetic mice[45,46]. Oxidized lipids can damage both lipids and proteins of membranes, ultimately destructing the integrity of organelles such as mitochondria. In consistent with the increased PUFA-CL to MUFA-CL ratio, we found that oxidized CL and acylcarnitine levels were increased in *Lcn2* KO BAT. As shown in Supplementary Fig. 7a, b, the levels of each of four measurable oxyl-cardiolipins as well as total oxyl-cardiolipins were significantly higher in *Lcn2* KO mice in the basal condition and had an increasing trend in the CL316, 243-treated condition. The levels of acylcarnitine (18:0, 18:1, 18:2), indicative of incomplete mitochondrial oxidation were significantly increased in *Lcn2* KO mice especially in the CL316, 243-treated condition (Supplementary Fig. 7c–e). These results suggest there exists mitochondrial damage and dysfunction in BAT of *Lcn2* KO mice, which may result from altered remodeling of cardiolipins. To support this, we showed that the MAM levels of DRP1, FACL4, and SigmaR1 were increased in response to CL316, 243 in WT mice, but this response was reduced leading to decreased levels of these proteins at the MAM in *Lcn2* KO mice (Supplementary Fig. 7f). Both MFN2 and FACL4 had decreased basal levels in *Lcn2* KO mice (Supplementary Fig. 7f).

It is known that damaged mitochondria can release their components also called mitochondrial DAMPs (mitochondrial damage-associated molecular pattern) triggering inflammatory responses. The DNA sensor cyclic GMP-AMP synthase (cGAS-STING) and the cytoplasmic NLR family pyrin domain containing 3 (NLRP3) are the two main pathways that can be activated by mitochondrial DAMPs. As one of the mitochondrial DAMPs, Cardiolipin can be externalized from the inner to the outer mitochondrial membrane where it gets oxidized and becomes proinflammatory leading to the activation of inflammasome under the mitochondrial stress[47,48]. Therefore, we assessed the activity of inflammatory pathways induced by mitochondrial DAMPs in *Lcn2* KO BAT. As illustrated in Supplementary Fig. 7g–j, the levels of STING and NLRP3 were significantly increased by CL316, 243 treatment in BAT of Lcn2 KO but not WT mice; the NLRP3 levels were significantly higher in Lcn2 KO BAT than WT BAT in the CL316, 243-treated condition. The levels of NFkB were significantly decreased by CL316, 243 in WT BAT but not in *Lcn2* KO BAT, leading to increased NFkB levels in *Lcn2* KO BAT upon CL316, 243 stimulation (Supplementary Fig. 7j). However, the ratio of p-NFkB to total NFkB was similarly increased in WT and *Lcn2* KO BAT (Supplementary Fig. 7k). These results support that *Lcn2* deficiency increases mitochondrial damage and its associated activation of inflammatory pathways.

Next, we assessed the expression of genes involved in mitochondrial function and oxidation. The heat map analysis showed that WT and *Lcn2* KO BAT had a distinct expression pattern of mitochondrial genes in response to CL316, 243 (Supplementary Fig. 8a). Most of the mitochondrial genes examined were upregulated by CL316, 243 in WT. However, this upregulation was diminished in *Lcn2* KO mice (Supplementary Fig. 8b–k), suggesting that *Lcn2* deficiency reduces the CL316, 243 induction of mitochondrial oxidation. To further prove mitochondrial dysfunction in *Lcn2* KO BAT, we measured ROS and mitochondrial membrane potential (MMP) in differentiated brown adipocytes. Compared to WT controls, *Lcn2* KO brown adipocytes had

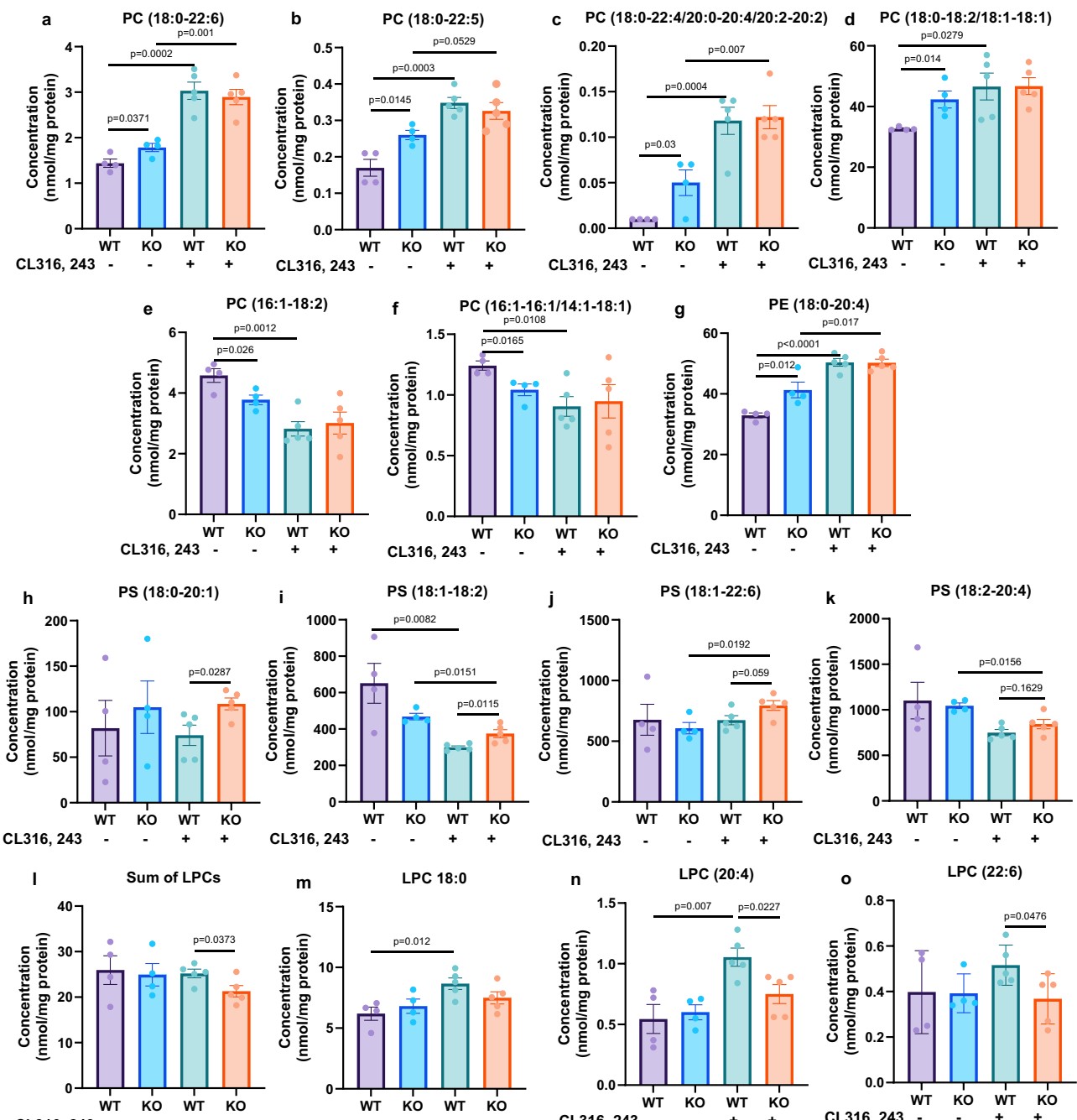

**Fig. 5 | Content of PC, PE, PS, and LPC species altered by *Lcn2* deficiency in BAT.** Individual altered PC (**a-f**), PE (**g**), and PS (**h-k**) species. Sum of three altered LPC species (**l**) and three individual LPC species (**m-o**) in BAT of WT and *Lcn2* KO mice treated with Saline or CL316, 243 for 14 days. Results are presented as mean ± SEM.

Student's *t*-test was performed to test differences between two independent groups. All tests were two-sided. *n* = 4 (saline-treated WT and *Lcn2* KO mice), *n* = 5 (CL316, 243-treated WT and *Lcn2* KO mice). Source data are provided as a Source data file.

significantly increased ROS but decreased MMP levels in the basal and treated conditions with CL316, 243 and inflammatory stimulation (Supplementary Fig. 9a–d). Moreover, the expression of antioxidant enzymes was decreased in either the basal or the CL316, 243-treated condition in *Lcn2* KO BAT (Supplementary Fig. 9e–j). The seahorse analysis of mitochondrial respiration in differentiated brown adipocytes demonstrated that *Lcn2* KO brown adipocytes had significantly decreased OCR as well as decreased maximal respiration and spare respiratory capacity compared to WT cells (Supplementary Fig. 9k–r). Together, these results further support that *Lcn2* deficiency reduces mitochondrial function.

**Lcn2 deficiency leads to compensatory activation of peroxisome function in BAT under the metabolic stimulation by CL316, 243**

The above results of increased LC-PUFA remodeling of phospholipids led us to investigate the impact of *Lcn2* deficiency on peroxisome function, as peroxisomes are the organelle that oxidize VLCFAs via β-oxidation and LC-PUFA metabolism[49]. Peroxisomes are also in contact with the ER, mitochondria, and lipid droplets and shared with the ER lipid biosynthesis pathways[49]. For example, peroxisomes are essential for the biosynthesis of plasmalogens, a major subclass of ether glycerophospholipids. Plasmalogens are generally enriched in the PUFAs, including DHA (C22:6) and arachidonic acid (C20:4). Plasmalogen

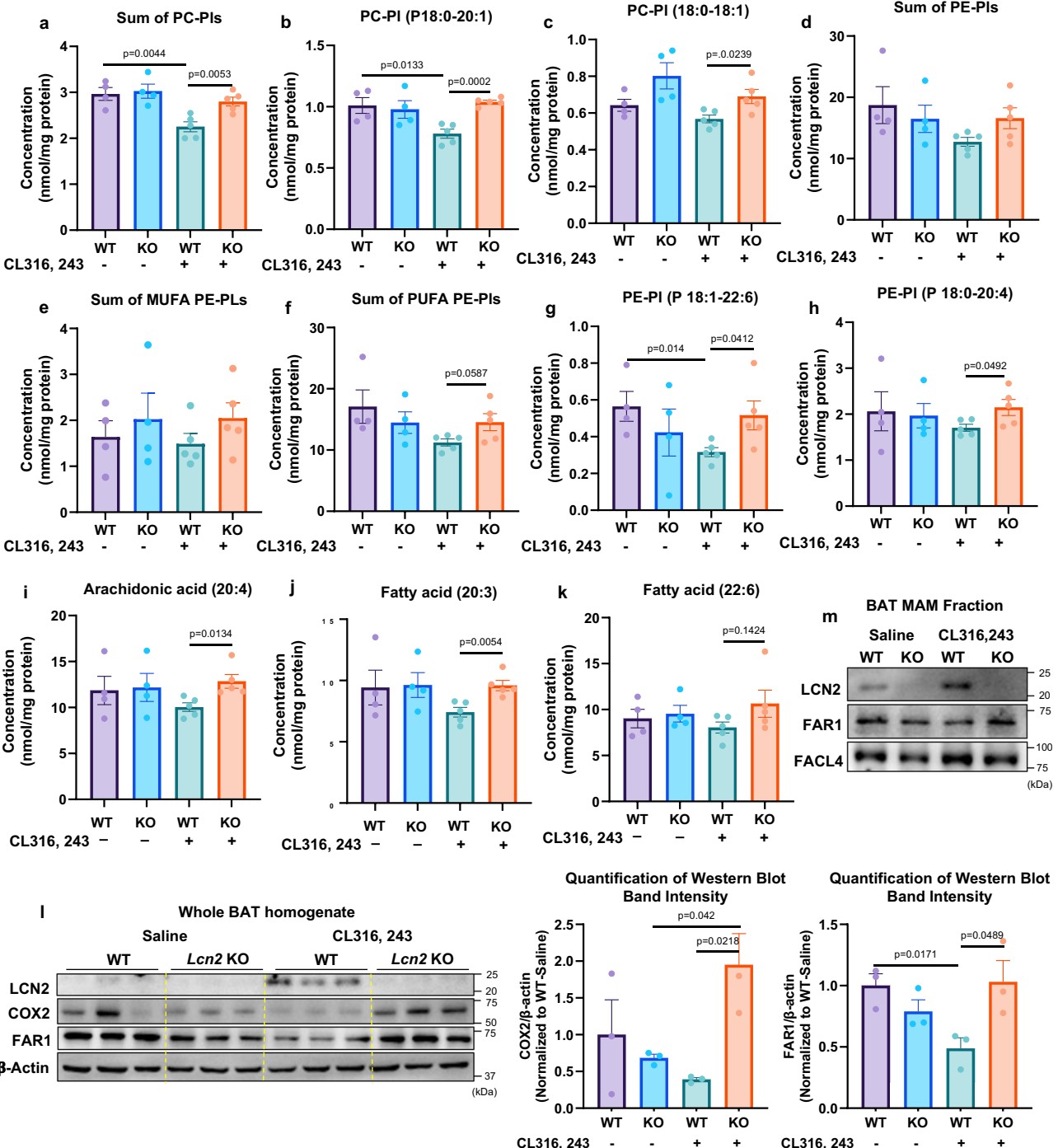

**Fig. 6 | Effect of *Lcn2* deficiency on plasmalogen biosynthesis, peroxisomal β-oxidation and LC-PUFA metabolism in BAT.** Sum of PC-Pls (**a**) and individual PC-Pls (**b**–**c**). Sum of PE-Pls (**d**), MUFA-PE-Pls (**e**), and PUFA-PE-Pls (**f**) as well as individual PE-Pls (**g**–**h**). The levels of LC-PUFA (**i**–**k**) in BAT of WT and *Lcn2* KO mice treated with Saline or CL316, 243 for 14 days. Protein expression of LCN2, COX2, and FAR1, a rate-limiting enzyme for plasmalogen biosynthesis in whole BAT homogenate (**l**) and isolated MAM fraction from BAT of male mice (**m**). MAM was isolated from pooled BAT of 4 male mice. Results are presented as mean ± SEM. Student's *t*-test was performed to test differences between two independent groups. All tests were two-sided. *n* = 4 (saline-treated WT and *Lcn2* KO mice), *n* = 5 (CL316, 243-treated WT and *Lcn2* KO mice). Source data are provided as a Source data file. Pl: plasmalogen.

synthesis initiates in peroxisomes and completes in the ER. More importantly, recent studies suggest that mitochondria and peroxisomes are connected physically and functionally[50] and the dysfunction of one organelle can be counteracted by activating the function of the other through a compensatory mechanism[50]. Interestingly, we found that peroxisome function remains higher levels of activation in *Lcn2* KO BAT compared to WT BAT in the CL316, 243-stimulated condition. For instance, the levels of PC-Pls, PE-Pls including sum of PE-Pls, PE-Pl

(P18:1-22:6), and PE-Pl (P18:0-20:4) were significantly higher in *Lcn2* KO BAT than WT BAT after CL316, 243 stimulation (Fig. 6a–h). This increased level is primarily due to the blunted response to the CL316, 243-induced reduction in PC-Pls and PE-Pls in *Lcn2* KO mice. We also found that *Lcn2* KO mice had increased levels of LC-PUFAs (C20:4, C20:3, and C22:6) in *Lcn2* KO BAT (Fig. 6i, j, k). Fatty acyl-CoA reductase 1 (FAR1) on peroxisomal membranes is a rate-limiting enzyme in plasmalogen biosynthesis, and FAR1 expression was upregulated

during ER stress in cardio myocytes[51]. We showed that LCN2 was upregulated in response to CL316, 243 in whole BAT homogenate as well as isolated MAMs (Fig. 6l, m). Interestingly, FAR1 was detected in MAMs (Fig. 6m) and had reduced protein levels upon CL316, 243 treatment in both whole BAT homogenate and MAMs from WT BAT (Fig. 6l, m). However, FAR1 levels were lower in the basal but significantly higher in the CL316, 243-treated condition in *Lcn2* KO BAT compared to WT controls (Fig. 6l, m). Increased COX2 levels also reflected higher AA (C20:4) levels in *Lcn2* KO BAT in response to CL316, 243 (Fig. 6l). Together, *Lcn2* KO mice have increased peroxisome function in response to CL316, 243 stimulation, leading to increased LC-PUFAs, which enter plasmalogen biosynthesis pathway due to increased FAR1 activity. The increased peroxisome function is reflective of a compensatory mechanism for mitochondrial dysfunction in *Lcn2* KO mice, and this occurs under the stimulation of fatty acid oxidation.

### *Lcn2* deficiency alters PA production and PA-regulated pathway activation in BAT

In addition to its role as an intermediate in phospholipid biosynthesis, PA also regulates the activity of phospholipid-metabolizing enzymes. PA production by diacylglycerol kinase (DGK) has been associated with the activities of enzymes PLD, PLA2, and LIPIN1 through altering PA levels[52–54]. PA also functions as signaling molecules required for the activation of signaling proteins, including phosphatidylinositol 4-phosphate 5-kinase[55,56], mammalian target of rapamycin (mTOR)[57], atypical isoforms of PKC[58], and PLCγ1[59]. Although the specific regulatory role of individual PA species remain unexplored, several PA species such as 16:0-18:1 and PA 14:0-16:1 have been paired with diacylglycerol kinase δ (DGKδ) and linked to type 2 diabetes[60,61]. DGKs convert DAG (diacylglycerol) to PA, playing an important role in the regulation of PA production and PA-regulated cellular function[62,63]. DGKζ is a negative regulator of PLD activity and PLD-catalyzed PA production[64]. Therefore, we determined how *Lcn2* deficiency affects DAG-PA production axis by measuring PA and DAG levels, phospholipid-metabolizing enzymes, and PA-regulated signaling proteins.

We were able to detect six PA species from isolated crude mitochondria (Supplementary Data 2), and the sum of these PA species was decreased in *Lcn2* KO BAT in the basal condition (Fig. 7a). CL316, 243 treatment led to a decrease in the sum of PA in WT BAT but an increase in *Lcn2* KO compared to WT BAT (Fig. 7a). Individually, the basal levels of PA (16:0-18:2), PA (16:0-18:1), and PA (18:0-20:2) were significantly lower in *Lcn2* KO BAT (Fig. 7b, c, e). CL316, 243 reduced PA (16:0-18:2) and PA (16:0-18:1) in WT BAT, but failed to do so in *Lcn2* KO BAT (Fig. 7b, c). However, CL316 243 treatment led to an increase in PA (14:0-16:1) and a trend towards an increase in PA (18:0-20:2) in *Lcn2* KO BAT compared to WT BAT (Fig. 7e, f). There was no change in PA (18:0-18:2) and PA (14:1-16:1) in *Lcn2* KO BAT (Fig. 7g, h). Additionally, we were able to detect 26 DAG species (Supplementary Data 2) and showed that the sum of 26 DAGs as well as unsaturated DAGs was significantly higher in *Lcn2* KO BAT in both basal and CL316, 243-treated conditions (Fig. 7h, j). CL316, 243 enhanced the total and unsaturated DAGs, but not saturated DAGs in *Lcn2* KO BAT (Fig. 7h–j). Further, four out of six examined DGK isoforms displayed significantly altered expression in *Lcn2* KO BAT (Fig. 7k–n). *Dgkε* and *Dgkη* had increased basal levels of mRNA expression in *Lcn2* KO BAT (Fig. 7k, l). CL316, 243 significantly upregulated the expression of *Dgkε*, *Dgkδ*, and *Dgkζ* in WT BAT but not in *Lcn2* KO BAT, which leads to relatively lower levels of these DGK isoforms in *Lcn2* KO BAT in the treated condition (Fig. 7l–n). These results suggest that *Lcn2* is required for the homeostasis of DAG-PA production and PA function.

To understand better how *Lcn2* deficiency links PA dysregulation to the disruption of phospholipid remodeling, we investigated the phospholipid-metabolizing enzymes. Interestingly, we found that PLD,

phospho-PLA2G4A, and PLA2G4A were lower in *Lcn2* KO BAT in the basal condition, but significantly higher compared to WT BAT after CL316, 243 treatment (Fig. 7o–s). LIPIN1 was higher in *Lcn2* KO BAT in both basal and CL316, 243-treated conditions (Fig. 7o–s). With regard to PA-regulated signaling pathways, increased PA function is known to activate mTOR[57,65,66] and MAPK/ERK signaling cascade[67–69] leading to decreased autophagy[70]. We showed that the protein levels of mTOR signaling components, including p70S6K, Rictor, and Raptor were increased, whereas LC3 levels were decreased in BAT of *Lcn2* KO mice with CL316, 243 treatment (Fig. 8a, d–g). We also observed that ERK phosphorylation was increased in *Lcn2* KO BAT compared to WT BAT after CL316, 243 treatment (Fig. 8b). Further, time-course treatment showed that CL316, 243 induced higher levels of phospho-ERK in differentiated *Lcn2* KO brown adipocytes in vitro (Fig. 8c). Collectively, we summarize the results and explain how *Lcn2* deficiency disrupts the acyl-chain remodeling of mitochondrial phospholipids in BAT via altering the recursive regulation of PA production and PA-producing/phospholipid-metabolizing enzymes as indicated in Fig. 8d.

## Discussion

In the present study, we have identified LCN2 as a PA binding protein and characterized its role in the regulation of the acyl-chain remodeling of mitochondrial phospholipids, peroxisomal LC-PUFA metabolism and plasmalogen biosynthesis, as well as mitochondrial dynamics and function in BAT. LCN2 exerts its function through regulating PA production and function, which recursively controls the activities of enzymes responsible for phospholipid metabolism and PA production. We have shown that LCN2 selectively binds to PA and is recruited to the MAM during inflammation and metabolic stimulation. Our results indicate that *Lcn2* deficiency increases the percentage of LC-PUFA-containing CLs, but decreases MUFA-containing CLs. *Lcn2* deficiency impairs mitochondrial fission-fusion balance, leading to mitochondrial fragmentation and dysfunction. We also found that *Lcn2* deficiency increases the peroxisome function as evidenced by increased LC-PUFA metabolism and plasmalogen biosynthesis to compensate mitochondrial dysfunction during metabolic stimulation of fatty acid oxidation.

We have demonstrated that *Lcn2* deficiency does not affect the total levels of CL and other phospholipids (PC, PE, and PS), but alters their fatty acid composition. For instance, LC-PUFA remodeling generally increases, whereas MUFA (16:1) remodeling decreases in *Lcn2* KO BAT, and this effect is universal, affecting all types of phospholipids, including CLs, PC, PE, PS, and ether phospholipids. This is likely caused by the global change that we have observed in the activity of multiple phospholipid-metabolizing and -remodeling enzymes rather than specific remodeling processes, as *Lcn2* deficiency does not alter either TAZ or ALCAT1. Due to the LC-PUFA remodeling of CLs, *Lcn2* deficiency damages the mitochondrial membrane and impairs mitochondrial dynamics, leading to mitochondrial dysfunction. This is evidenced by the increased levels of oxidized CL and acyl-carnitine, the reduced OCR/maximal respiration, the downregulation of mitochondrial genes, and the increased activation of mitochondrial DAMPs−induced inflammatory pathways, including cGAS-STING and inflammasome in *Lcn2* KO BAT. Decreased mitochondrial function triggers a compensatory increase in peroxisome function and LC-PUFA metabolism, leading to increased levels of LC-PUFA (C20:4, 20:3, and C22:6) and increased biosynthesis of LC-PUFA-containing plasmalogens in *Lcn2* KO BAT. Additionally, our data of increased ROS levels and decreased MMP associates *Lcn2* deficiency-induced oxidative stress with mitochondrial dysfunction.

Further, we found that *Lcn2* deficiency significantly alters DAG-PA production and PA/DAG levels. These changes correlate well with phospholipid-metabolizing enzymes that regulate PA and DAG levels, including PLD, PLA2, LIPIN1, and DGKs in both basal and CL-316, 243-treated conditions (Fig. 8d). Numerous studies have demonstrated

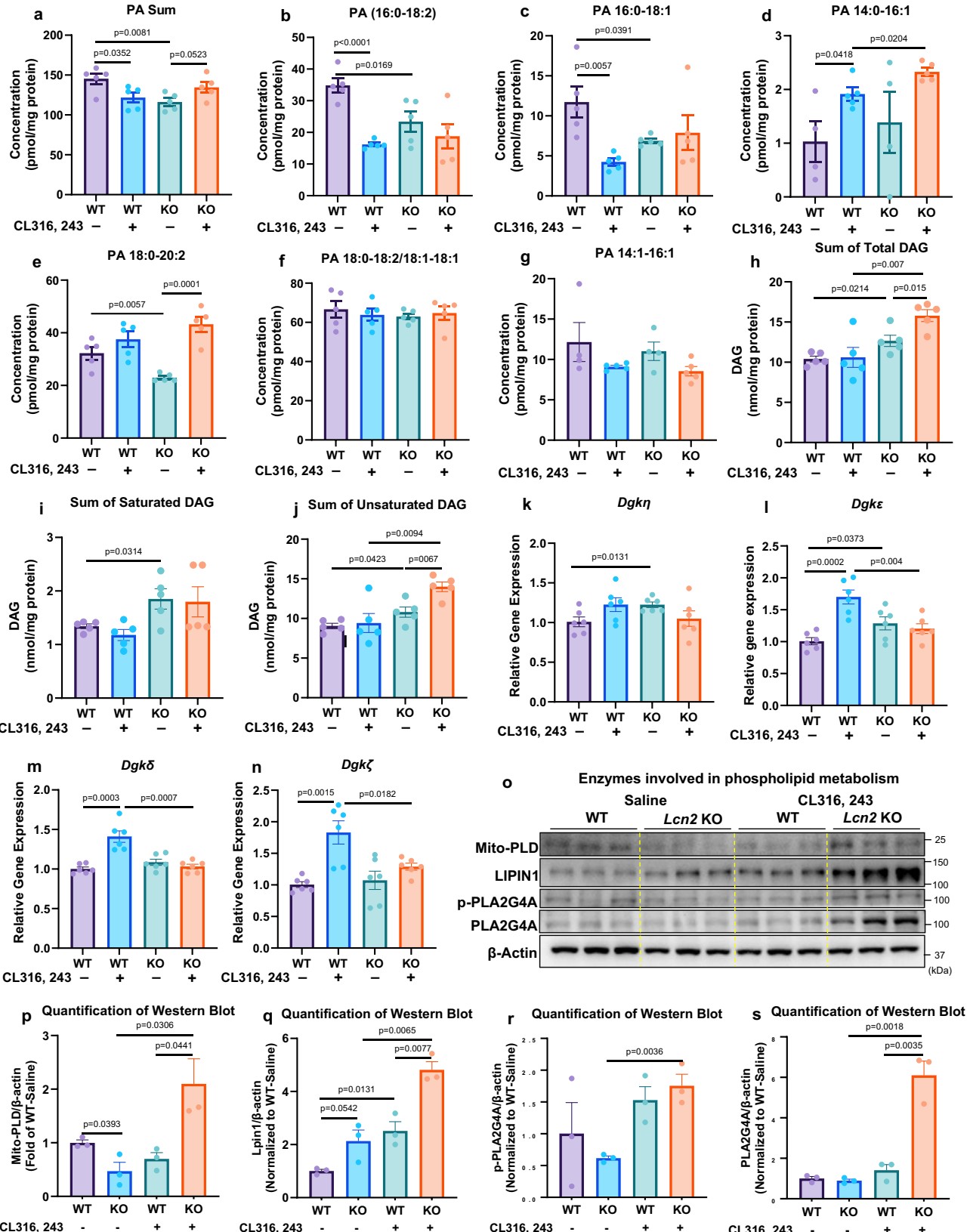

**Fig. 7 | Effect of *Lcn2* deficiency on DAG and PA production and phospholipid-metabolizing enzymes in BAT.** The levels of measurable PA (**a**–**g**) species and DAG species (**h**–**j**); *n* = 5 per group. Gene expression of DGK isoforms (**k**–**n**); *n* = 6 per group. Protein expression levels of phospholipid-metabolizing enzymes (**o**) in BAT homogenate of WT and *Lcn2* KO mice treated with Saline or CL316, 243 for 14 days. Quantification of western-blotting band intensity (**p**–**s**) (*n* = 3 mice). Results are

presented as mean ± SEM. Student's *t*-test was performed to test differences between two independent groups. All tests were two-sided. The animal experiment with CL316, 243 treatment was repeated 3 times independently; *n* = 5–6 mice per group for each independent experiment. Source data are provided as a Source data file.

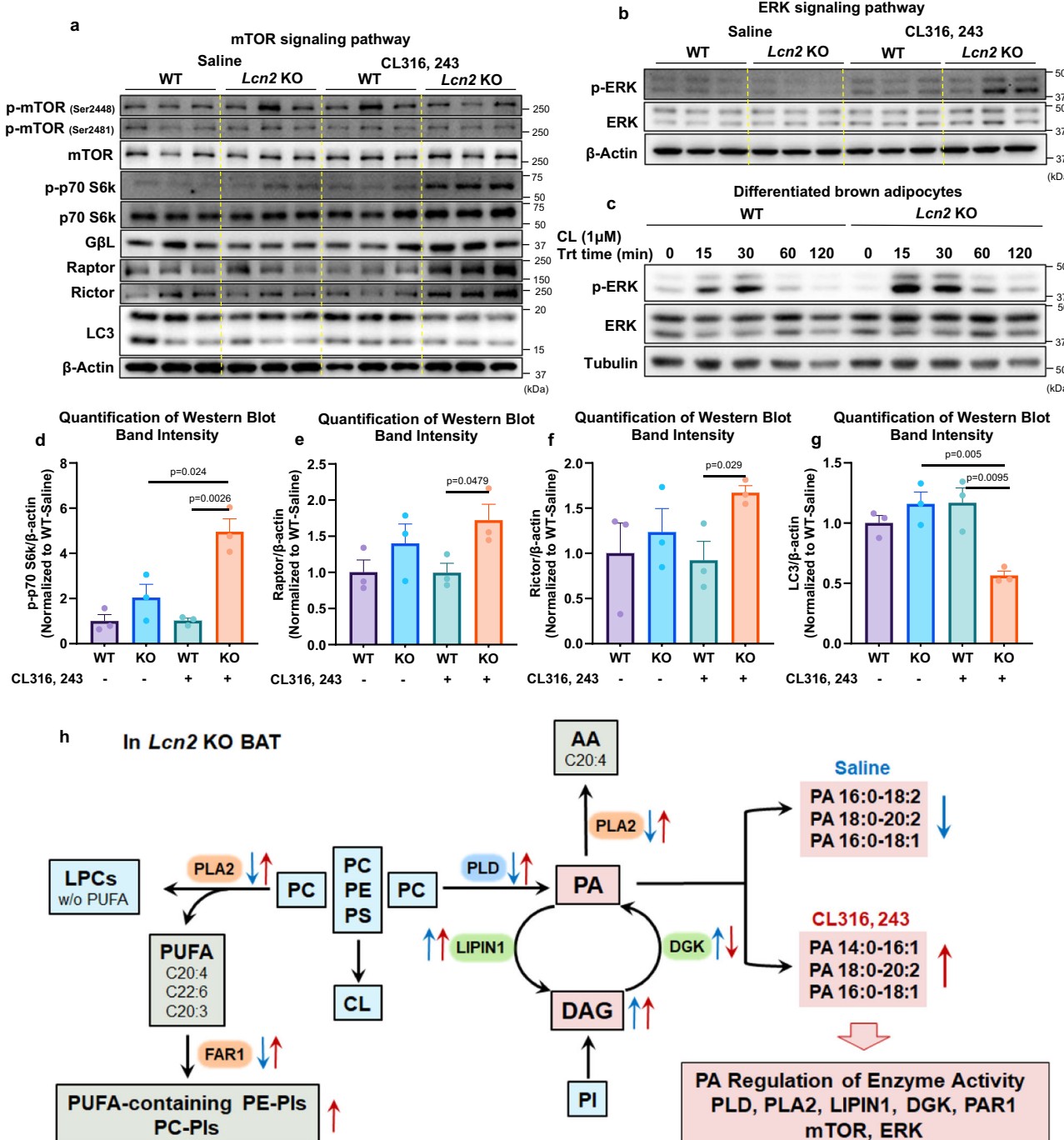

**Fig. 8 | Effect of *Lcn2* deficiency on PA-regulated signaling pathways in BAT.** Protein expression levels of mTOR signaling components (**a**) and ERK (**b**) in BAT homogenate of WT and *Lcn2* KO mice treated with Saline or CL316, 243 for 14 days. Time-course of ERK protein induction by CL316, 243 in differentiated brown adipocytes (**c**). Cell culture experiments were repeated twice independently. (**d**–**g**) quantification of western-blotting band intensity (*n* = 3 mice). Results are presented as mean ± SEM. Student's *t*-test was performed to test differences between two independent groups. All tests were two-sided. The diagram depicts the disrupted processes of acyl-chain remodeling and metabolism of phospholipids and the altered signaling PA production and function in *Lcn2* KO BAT (**h**). Blue arrows indicate changes in enzymes and lipids in the basal condition. Red arrows represent changes in the CL316, 243-treated condition. Source data are provided as a Source data file.

that PA production is essential for the activation of these enzymes[52–54], suggesting a reciprocal relationship between PA production and phospholipid-metabolizing enzyme activation. Intriguingly, we have found that changes in the levels of PA species particularly PA 18:0-20:2 and PA 16:0-18:1 correlate well with the activity of PLD, PLA2, and FAR1, suggesting that LCN2 is a critical regulator of PLD, PLA2, and FAR1 via regulating the production and function of these two PA species. As indicated in Fig. 8d, in the absence of *Lcn2*, PLD and PLA2 activity is decreased in the basal condition, promoting the conversion of more LC-PUFA-containing PC/PE into LC-PUFA-containing CLs. After CL316, 243 treatment, the levels of PA 18:0-20:2 and PA 16:0-18:1 plus PA 14:0-16:1 are increased, which also pairs with increased activities of PLD, PLA2, and FAR1. This increase could cause the increased release of LC-PUFA from PC (mostly) and the decreased levels of LPC-containing LC-PUFA (C20:4 and C22:6). Along with the increased release of LC-PUFA, increased FAR1 drives the peroxisomal biosynthesis of LC-PUFA-

containing plasmalogens in *Lcn2* KO BAT. Due to the technical challenge of measuring low levels of PA in isolated mitochondria, it is likely there are more PA species that were not detected in this study and that their levels and activity are also regulated by LCN2. However, our results clearly indicate that LCN2 plays a critical role in phospholipid metabolism and remodeling through controlling PA production and function, thereby regulating phospholipid-metabolizing enzymes. Future studies are warranted to determine the regulatory role of LCN2 in specific PA species as well as to identify proteins associated with LCN2 and their functions in phospholipid metabolism.

At the molecular level, PA can interact with various proteins to modulate their catalytic activity and/or their membrane association[71]. Specifically, PA has been reported to directly activate mTOR and ERK pathways[57,65–67,69,72]. PA together with saturated phospholipids suppress the constriction of DRP1 oligomerization that is required in mitochondrial division[73]. Similar to LCN2, DRP1, and MNF2, mTOR and ERK are found in mitochondria and MAM[74]. Therefore, we hypothesize that *Lcn2* deficiency has an impact on the activation of mTOR and ERK. As we expected, *Lcn2* deficiency exacerbates the activation of both mTORC1 and ERK pathways and decreases autophagy activation in the CL316, 243-treated condition under which PA levels are increased. This data further supports the role of LCN2 in the regulation of metabolic pathways that are controlled by PA signaling. Specifically, LCN2 may serve as a negative regulator of mTOR and ERK activation through suppressing PA production and function.

In summary, we have demonstrated that LCN2 plays a critical role as a PA-binding protein in the homeostatic regulation of PA production and function, by which LCN2 regulates the LC-PUFA remodeling of phospholipids, the peroxisomal LC-PUFA metabolism and plasmalogen biosynthesis, as well as mitochondrial dynamics and function. As indicated in a graphic summary (Supplementary Fig. 10), *Lcn2* deficiency disrupts signaling PA activity through affecting PA production and function. This, in turn, deregulates the phospholipid-metabolizing enzymes that require PA for activation. Consequently, this disruption impairs the remodeling of mitochondrial phospholipid CL, resulting in an increased ratio of PUFA- to MUFA-containing CLs and CL oxidation, ultimately leading to mitochondrial damage and dysfunction. The PA activation by *Lcn2* deficiency also triggers the mTOR signaling pathway, further contributing to mitochondrial dysfunction and oxidative stress. This, in turn, leads to compensatory peroxisome activation and increased biosynthesis of LC-PUFA-containing plasmalogens. Changes in LC-PUFA levels also affects the production of lipid ligands necessary for the activation of transcriptional factors such as PPARα, which contributes to mitochondrial oxidative metabolism. Finally, dysfunctional mitochondria fail to maintain redox balance, leading to increased ROS levels and oxidative stress under inflammatory and metabolic stimulation. However, further investigations are warranted to explore the role of LCN2 in mitochondria-peroxisome communication and whether PA serves as a signaling message connecting LCN2-regulated functional compensation between these two organelles during metabolic stress.

## Methods

### Mice
*Lcn2*-kockout (KO) mice in C57BL6/J background were kindly provided by Dr. Alan Aderem, Institute for Systems Biology, Seattle, Washington, USA, where they were originally generated by Dr. Shizuo Akira, Research Institute for Microbial Diseases, Osaka University, Japan. C57BL6/J mice at 8 weeks of age were purchased from the Jackson Laboratory. Heterozygous mating scheme was used to generate WT and *Lcn2* KO mice as previously described[22]. *Lcn2* KO mice were backcrossed onto the C57BL6/J background for 10 generations before mice were used for the experiments. Animals were housed at 22 °C in a specific pathogen-free facility (12-h light/dark cycle, 60–70% humidity) at the University of Minnesota and fed a regular chow diet (Teklad

#2018,18% calories from fat, 24% calories from protein, 58% calories from carbohydrate, 3.1 kcal/g, Harlan Laboratories, Madison, WI, USA), with free access to water. Animal studies were conducted with the approval of the University of Minnesota Animal Care and Use Committee and conformed to the National Institute of Health guidelines for laboratory animal care (IACUC 2102A38852). Male mice were used for all the experiments in this study. For acute cold exposure, WT and *Lcn2* KO male mice were fasted overnight prior to being placed in individual cages at 4 °C with free access to water. For the study with lipopolysaccharide (LPS, #L4391, Sigma-Aldrich, St. Louis, MO, USA) stimulation, WT male mice at 8-10 weeks of age were given 0.3 mg/kg of LPS via intraperitoneal injection. After 6 h of treatment, mice were sacrificed for blood and tissue collection. The age-matched mice with i.p. injection of saline served as controls. For the study with the treatment of CL316,243 (#L4391, Sigma-Aldrich), WT and *Lcn2* KO male mice at 8–10 weeks of age were given either saline or CL316,243 (0.5 mg/kg BW) via i.p. injection once a day for 14 days. At the end of the experiments, mice were sacrificed under the fed condition. Blood and tissue samples were collected, weighed, and stored at −80 °C for further analysis.

### 3T3-L1 cell culture
3T3-L1 cells were purchased from the American Type Culture Collection (ATCC, #CL-173, Manassas, VA). 3T3-L1 cells were grown in DMEM (#10-017-CV, Corning, NY, USA) with 10% bovine calf serum (#12133 C, Sigma-Aldrich) and 100 IU/ml penicillin-streptomycin (#15140122, Gibco, USA) until confluence. Cells were then induced to differentiate into adipocytes with the differentiation cocktail containing DMEM, 10% fetal bovine serum (#10082147, Gibco Sigma), 100 IU/ml penicillin/streptomycin, 115 g/ml 3-Isobutyl-1-methylxanthine (#I5879, Sigma-Aldrich), 1 μg/ml insulin (#I6634, Sigma-Aldrich Sigma), 100 ng/ml dexamethasone (#D8893, Sigma-Aldrich), and 125 M indomethacin (#I7378, Sigma-Aldrich) for 2 days. The cultures were then continued with DMEM with 100 IU/ml penicillin-streptomycin, 10% fetal bovine serum, and 1 μg/ml insulin for 6 days. Both 3T3-L1 preadipocytes and adipocytes were used for co-localization of LCN2 with ER or mitochondrial markers as described in the confocal microscope section.

### Cell culture and differentiation of primary stromal-vascular cells
Stromal-vascular (SV) cells were isolated from brown adipose tissue (BAT) or inguinal white adipose tissue (WAT) of WT and *Lcn2* KO male mice as previously described[21,24]. Briefly, primary SV cells were isolated from adipose tissue and used for adipocyte differentiation. Brown adipose tissue was removed from WT and *Lcn2* KO male mice at 10-12 weeks of age, minced, and digested with Krebs-Ringer bicarbonate HEPES buffer (KRBH, 120 mM NaCl, 4 mM KH$_2$PO$_4$, 1 mM MgSO$_4$, 1 mM CaCl$_2$, 10 mM NaHCO$_3$, 30 mM HEPES, pH 7.4) containing 1% bovine serum albumin (BSA, #A1311, US Biological, USA), 100 μg/ml Gentamycin Sulfate (IB02030, IBI Scientific, Dubuque, IA, USA), and 5 mg/ml collagenase Type 1 (#LS004197, Worthington Biochemical Corporation, New Jersey, USA). After 1.5 h digestion, SV cells were separated from floating adipocytes through centrifugation at 1200 rpm for 10 min and washed with KRBH buffer twice. After the final wash, SV cells were plated at the same cell density on 6-well plates and cultured in DMEM containing 20% fetal bovine serum and 100 IU/ml penicillin/streptomycin until full confluence. Cells were then treated with the differentiation cocktail consisting of DMEM, 10% fetal bovine serum, 100 IU/ml penicillin/streptomycin, 115 g/ml 3-Isobutyl-1-methylxanthine, 1 μg/ml insulin, 100 ng/ml dexamethasone, 125 M indomethacin, and 20μM L-3,3′,5-Triiodothyronine (#642511, Sigma-Aldrich). Three days later, the differentiation cocktail was replaced with DMEM containing 10% fetal bovine serum, 100 IU/ml penicillin/streptomycin, and 1 μg/ml insulin, and cells were cultured for an additional 6 days. On day 7–9 of differentiation, differentiated WT and *Lcn2* KO brown adipocytes were treated with Earle's balanced salt

solution (EBSS, #24010043, Gibco) for 1 h, followed by the treatment with high glucose (25 mM) DMED containing 0.3 mM palmitic acid (#P0500, Sigma-Aldrich) for 1 h, 3 h and 6 h, respectively. The whole cell lysates were collected for further analysis.

## Quantitative real-time PCR

Total RNA was isolated from tissues using TRIZOL reagent (#15596018, Invitrogen, Carlsbad, CA, USA). cDNA was synthesized using High-Capacity cDNA Reverse Transcription Kit (#4368814, Applied Biosystems, CA, USA). Real-time quantitative PCR was conducted using PowerUp SYBR Green Master Mix for qPCR (#A25742, Applied Biosystems) with a QuantStudio™ 3 Real-time PCR System (Applied Biosystem). The ΔΔCt method was used to calculate mRNA expression and β-actin or TBP1 served as an internal control. The primer sequences for amplifying the target genes are summarized in Supplementary Table 1.

## Western blot analysis

Equivalent amounts of protein were run on an SDS-PAGE gel and transferred to a nitrocellulose membrane (0.45 μm, #1620115, Bio-Rad, Hercules, CA, USA). After blocking with 5% nonfat dry milk (#M0842, Labscientific, Bethesda, MD, USA) in TBS containing 0.1% Tween-20 (#P9416, Sigma-Aldrich), the membrane was incubated with the primary antibody overnight at 4 °C. After three washes with PBST, the membrane was incubated with secondary antibodies (Anti-rabbit, Anti-mouse, Anti-rat or Anti-goat HRP secondary antibody, R&D Systems) for 1 h at room temperature. The immunoreactive protein bands were visualized by SuperSignal West Pico PLUS Chemiluminescent Substrate (#34577, ThermoFisher Scientific, Waltham, USA) or SuperSignal West Atto Ultimate Sensitivity Chemiluminescent Substrate (#38556, ThermoFisher Scientific) and scanned using the iBright Imaging System (Thermo Fisher Scientific). The primary antibodies used in this study include anti-DRP1, OPA1, Mitofusin-2, β-Actin, Phospho-DRP1 (Ser616), SigmaR1, cGAS, STING, NLRP3, p-NF-κB p65 (Ser536), NF-κB, LIPIN1, COX2, p70 S6K, p-p70 S6K, p-ERK, ERK, LC3, α/β-Tubulin, Phospho-mTOR (Ser2481), mTOR, Raptor, Rictor, GβL, and Phospho-mTOR (Ser2448) (Cell Signaling Technology, Beverly, MA, USA), Lipocalin 2 (R&D Systems, Minneapolis, MN, USA), FACL4 (Novus Biologicals, Centennial, CO, USA), Mito-PLD (MBL, Nagoya, Japan), FAR1, p-PLA2G4A and PLA2G4A (AB colonal, Woburn, MA, USA), CRLS1 (Proteintech, China), TAZ (Santa Cruz, USA). Anti-ALCAT1 antibody was kindly gifted by Dr. Yuguang Shi. The sources of primary and secondary antibodies as well as the antibody dilution are provided in Supplementary Table 2.

## Membrane lipid binding assay

In the lipid-binding assay, membrane lipid strips (#P-6002, Echelon Bioscience, Salt Lake City, USA) spotted with indicated lipids were incubated with blocking buffer PBS-T containing 0.1% v/v Tween-20 and 3% BSA overnight at 4 °C, then with 4 μg/ml of mouse recombinant LCN2 (#1857-LC-050, R&D systems) for 1 h, followed by anti-LCN2 antibody (#AF1857, R&D Systems) in PBS-T with 3% BSA for 1 h at room temperature (RT). After washing for three times, the membrane was incubated with an anti-goat HRP antibody (#HAF019, R&D Systems) for 1 h at RT. The bound LCN2 was detected and visualized using SuperSignal West Atto Ultimate Sensitivity Chemiluminescent Substrate and scanned using the iBright Imaging System (Thermo Fisher Scientific).

## Protein pull-down assay with lipid-coated beads

LCN2 protein pull-down assay was performed with phosphatidic acid (PA)-coated beads (#P-B0PA; Echelon Biosciences, Salt Lake City, USA) according to the manufacturer's instructions. For pull-down assay, 50 μl PA beads were incubated with 0.1 μg, 1 μg, 3 μg and 6 μg of recombinant mouse LCN2 protein, respectively in a binding buffer (PBS with 0.1% Tween-20 and 0.5% fatty acid-free bovine-serum albumin, (BSA #03117057001, Roche, Germany)). After 3 h of incubation at 37 °C with rotation, beads were washed five times with wash buffer (PBS with 0.1% Tween-20). Bound proteins were eluted by adding an equal volume of 2X Laemmli sample buffer (#161-0737, Bio-Rad). Following heating at 70 °C for 10 min, the bound proteins were analyzed by Western blotting. Control beads (#P-B000, Echelon Biosciences) were used as a negative control to check the background binding.

## Confocal microscopy

Confocal microscope was performed with Nikon A1 confocal microscope with a 60× oil- immersion objective at the University of Minnesota Imaging Center using identical imaging parameters. Double immunostaining was conducted to examine the co-localization of LCN2 with ER and mitochondrial marker. 3T3-L preadipocytes and adipocytes at day 5 of differentiation were treated with LPS (1 μg/ml) for 4 h, followed by double immunostaining with antibodies against LCN2 (1:40 dilution, #AF1857, R&D Systems) and calnexin (1:100 dilution, #ab75801, Abcam), an ER marker or with antibodies against LCN2 (1:40 dilution) and TOM 20 (1:50 dilution, #sc-11415, Santa Cruz), a mitochondrial marker. Images were collected in channel series and Z series manner, giving the 3D image datasets. Colocalization was quantitatively analyzed by Fiji-JACoP, and Pearson's coefficients were calculated based on 3D image datasets.

For the morphology of mitochondrial fusion and fission, differentiated WT and Lcn2 KO brown adipocytes were treated with EBSS or HG + PA for 3 h. The cells were then rinsed with PBS and incubated in prewarmed staining solution containing 100 nM MitoTracker Deep Red FM (#M22426, Invitrogen) at 37 °C for 30 min. After the staining with MitoTracker, the cells were fixed with 4% formaldehyde for 20 min, followed by the incubation with 10 μM DAPI (#D9542, Sigma-Aldrich) at 37 °C for 15 min. The images were captured by Nikon A1 confocal microscope.

## Lipidomics

For lipidomic analyses, 50 mg of frozen BAT were placed in a glass Teflon homogenizer (RW20, IKA, Germany) and homogenized (7 strokes at 20 setting) in isolation buffer (20 mM Tris/HCl, 220 mM Mannitol, 70 mM Sucrose, 1 mM EDTA, 0.1 mM EGTA, 0.1% BSA, pH 7.4). The homogenate was transferred to 50 mL conical tube and centrifuged at 700 g at 4 °C for 10 min. The supernatant was transferred to 50 mL polycarbonate round bottom centrifuge tube (avoiding lipid layer at the top), filtered through sterile gauze during transfer, and then centrifuged at 9000 g, 4 °C for 10 min. After removing the supernatant, the pellet was re-suspended in the appropriate amount of assay buffer for lipidomics.

Lipid species were analyzed using multidimensional mass spectrometry-based shotgun lipidomics approach[75]. In brief, each sample was homogenized and an equivalent of 0.08 mg protein homogenate determined by BCA protein assay kit (#23227, ThermoFisher Scientific) was added to a glass tube along with pre-mixed lipid internal standards. Then, lipid extraction was performed using a modified Bligh and Dyer approach[76]. Finally, the lipid extract was dispersed in chloroform:methanol (1:1, v-v) at a ratio of 800 μL/mg protein for storage. For shotgun lipidomics, the lipid extract was further diluted to a total lipid concentration of ~2 pmol/μL. The mass spectrometric analysis was performed on a triple quadrupole mass spectrometer (TSQ Altis, Thermo Fisher Scientific, San Jose, CA) and a hybrid quadrupole-Orbitrap mass spectrometer (Q-Exactive, Thermo Fisher Scientific, San Jose, CA), both equipped with an automated nanospray ion source device (TriVersa NanoMate, Advion Bioscience Ltd., Ithaca, NY) as described previously[77]. The data processing and analysis was performed based on the principles of shotgun lipidomics such as ion peak selection, baseline correction, isotope effect correction, data transfer, peak intensity comparison, and quantitation.[75,78,79].

Ionization voltages of −1.1, −0.95, and +1.2 kV and gas pressures of 0.3, 0.15, and 0.3 psi were employed on the nanomate apparatus for the analyses of anionic lipids, PE, and PC, respectively. The first and third quadrupoles were used as independent mass analyzers with a mass resolution setting of 0.7 Thomson, whereas the second quadrupole served as a collision cell for tandem mass spectrometry (MS/MS) as previously described[80]. The final lipidomics results were normalized to the protein content (i.e., nmol/mg protein).

## Transmission electron microscopy (TEM)

Mice were anesthetized and perfused with 2% glutaraldehyde and 2% paraformaldehyde in 0.1 M phosphate buffer. Brown adipose tissue was then dissected and cut into small pieces and fixed overnight at 4 °C in 2% glutaraldehyde and 2% paraformaldehyde in 0.1 M phosphate buffer. Next day the tissues were rinsed in phosphate buffer and fixed in 1% osmium tetroxide and 2% uranyl acetate for 2 hours each. The samples were then dehydrated in a graded ethanol series (50%, 70%, 85%, 95%, 100% × 2) for 10 min in each step. Tissues were washed with acetone 3 times for 15 min and infiltrated with Embed resin at the University of Minnesota Imaging Center. Samples were stained with uranyl acetate/lead citrate and high-resolution images were acquired with a JEOL JEM-1400plus electron microscope (JEOL). Mitochondrial size were analyzed using Image J (Version: 2.3.0). Briefly, individual mitochondrial area in each TEM image was measured using a freehand option. A total of 496 WT and 510 *Lcn2* KO mitochondria from 4-5 BAT tissue sections on a TEM grid per mouse of 3 WT and 3 *Lcn2* KO mice were measured and analyzed.

## BAT subcellular fractionation and MAM isolation

A previously published protocol was followed to purify MAM fraction[25]. Briefly, after homogenization of 100 mg of BAT with a Teflon potter in Isolation Buffer (225 mM mannitol, 75 mM sucrose, 0.5% BSA, 0.5 mM EGTA and 30 mM Tris-HCl, pH 7.4), cellular debris and nucleus were removed with two centrifugations at 740 x g for 5 min. The supernatant was collected and then centrifuged at 9,000 x g for 10 min, and the pellet was resuspended in Mitochondria Buffer (MB) (250 mM mannitol, 5 mM HEPES and 0.5 mM EGTA, pH 7.4). Pure MAM fractions were obtained from the crude mitochondria fraction with a Percoll medium (#P1644, Sigma-Aldrich) centrifugation at 95,000 x g for 30 min in a SW40 rotor (Beckman). Pure MAMs were collected from a white band in the middle of the tube. They were then diluted in MB and centrifuged at 6,300 x g for 10 min to remove mitochondrial contamination, pelleted with a 1-h centrifugation at 100,000 x g in a 70Ti rotor (Beckman), and finally resuspended in MB. To estimate the amount of each fraction within the BAT, proteins were determined using the BCA Protein Assay kit.

## Reactive oxygen species (ROS) and mitochondrial membrane potential (MMP) assay

ROS and MMP in differentiated brown adipocytes were measured using ROS Detection Cell-Based Assay Kit (#601520, Cayman Chemical, Ann Arbor, USA) and TMRE MMP Assay Kit (#701310, Cayman Chemical, Ann Arbor, USA), respectively, following the manufacturer's instructions. Briefly, primary SV cells isolated from BAT were seeded into black 96-well plates (#3603, Corning costar) at a concentration of $6 \times 10^3$ cells/well, followed by 48 h incubation. After reaching the full confluence, brown SV cells were induced to differentiate into adipocytes as described above. At the day 7 of differentiation, WT and *Lcn2* KO differentiated brown adipocytes were treated without or with CL 316,243 (1 µM), LPS (1 µg/ml), or IL-1β (1 ng/ml, # RMIL1BI, Invitrogen) for 12 h. After incubating with 10 µM DCFDA for ROS measurement or 50 nM TMRE for MMP measurement for 30 min at 37 °C, cells were washed with the cell-base assay buffer for 3 times and then subjected to the measurement of DCFDA (with 488 nm excitation and 530 nm emission wavelength) or TMRE (with 530 nm excitation and 580 nm emission wavelength) fluorescence intensities by Tecan infinite 200Pro microplate reader (TECAN, Germany).

## Hematoxylin and eosin staining of tissues

Tissues were fixed in 10% neutral buffered formalin (#89370-094, VWR, Germany) for 3-5 days, then dehydrated by ethanol solutions and processed for embedding in paraffin. Tissue samples were H&E stained using a standard protocol at the University of Minnesota Histology Core. Briefly, after deparaffinization and rehydration, tissues were sectioned with 5–6 µm thickness and stained in Hematoxylin for one minute and then rinsed with distilled water. After hematoxylin staining, the tissues were counterstained with Eosin solution for one minute, followed by dehydration through 95% EtOH and 100% EtOH and Xylene clearance. At last, the tissue sections were mounted with resinous mounting medium. Images were captured using a Leica microscope.

## Oxygen consumption rate (OCR)

OCR of differentiated brown adipocytes was determined using Seahorse XF Cell Mito Stress Test Kit (#103015-100, Agilent Technologies, Santa Clara, CA, USA) by a Seahorse XFe96 Analyzer (Agilent Technologies) according to the manufacturer's instruction. Briefly, WT and *Lcn2* KO brown adipocytes were cultured in XF96 microplates. Brown adipocytes were cultured in Seahorse XF DMEM medium (#103575-100, Agilent Technologies) containing 10 mM glucose, 2 mM L-glutamine, and 2 mM sodium pyruvate and treated with CL 316,243 (1 µM) for 3 h, 6 h, and 12 h, respectively. Firstly, 1 µM oligomycin was added to inhibit ATP synthase. Secondly, maximal respiration rate was determined by adding 2 µM FCCP. Lastly, mitochondrial respiration was shut down by the addition of 0.5 µM Rotenone/antimycin A. Brown adipocytes were labeled with Hoechst 33342 (#AS-83218, AnaSpec, Fremont, CA, USA) and counted with a Cyation 1 Cell Imaging Multi-Mode Reader (BioTek, Winkowski, VT, USA). Oxygen consumption rates were normalized by brown adipocyte number.

## Serum fatty acids and insulin

Mouse serum were prepared by centrifuging the blood samples at 2,000 x g for 10 min in a refrigerated centrifuge. Free fatty acids levels in serum were determined using free fatty acid quantification kit (#MAK044, Sigma-Aldrich) and serum insulin levels were measured using mouse insulin ELISA kit (#EMINS, Invitrogen) following manufacturer's instructions.

## Statistical analysis

Variables are presented as mean ± SEM and one-way ANOVA with Tukey's multiple comparison test was used to assess differences for multiple groups. Student's *t* test was used to test differences between two independent groups with GraphPad Prism 10. All tests were two-sided. General linear models analysis was conducted to determine genotype and treatment interaction. Principal components analysis (PCA) was performed and plotted using GraphPad Prism 10. Heatmaps were plotted using Clustvis with row sacling[81]. Hierarchical clustering using correlation distance and complete linkage was performed. All testing used a 2-sided alpha level of 0.05. All statistical analyses were performed with SAS software version 9.4 (SAS Institute, Cary, North Carolina) or GraphPad Prism 10.

## Reporting summary

Further information on research design is available in the Nature Portfolio Reporting Summary linked to this article.

# Data availability

Data are available within the Article or information. The datasets generated and/or analyzed during the current study, which have not been deposited in a public repository can be obtained from the

corresponding author upon request. Source data are included with this paper. Source data are provided with this paper.

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

## Acknowledgements

We thank Dr. Yuguang Shi from University of Texas Health Science Center at San Antonio for kindly providing anti-ALCAT1 antibody and assistance with the western blotting analysis of ALCAT1. This work was supported by NIDDK Grant (R01 DK123042) awarded to X.C. and partially supported by NIA Grant (RF1 AG061872) awarded to X.H. The Functional Lipidomics Core at Barshop Institute for Longevity and Aging Studies was partially supported by NIA P30 AG044271 and P30 AG013319. This work was also supported by the resources and staff at the University of Minnesota University Imaging Centers (UIC). SCR_020997.

## Author contributions

H.S., H.G., X.Q., and T.L. performed experiments and analyzed data. X.C. and H.S. analyzed the lipidomics data. C.Q. performed the lipidomic analysis. X.H. supervised the lipidomics work, provided insightful comments on the interpretation of lipidomics data, and edited the manuscript. G.C. performed the transmission electron microscopy (TEM). M.S. supervised the TEM work and the quantitative confocal data analysis. H.S. performed the quantitative TEM data analysis. P.Y. performed the membrane-lipid binding assay. D.A.B. supervised the membrane-lipid binding assay, conceptualized the project, and interpreted the data. X.C. conceptualized the project, supervised experiments, analyzed and interpreted the data. X.C. and H.S. wrote the manuscript.

## Competing interests

The authors declare no competing interests.

## Additional information

[1]Department of Food Science and Nutrition, University of Minnesota-Twin Cities, St. Paul, MN 55108, USA. [2]Barshop Institute for Longevity and Aging Studies, Department of Medicine, University of Texas Health Science Center at San Antonio, San Antonio, TX 78229-3900, USA. [3]University Imaging Centers, University of Minnesota-Twin Cities, Minneapolis, MN 55455, USA. [4]Department of Biochemistry, Molecular Biology and Biophysics, University of Minnesota-Twin Cities, Minneapolis, MN 55455, USA. ✉e-mail: xlchen@umn.edu

