## [Peer Review File · Nature Communications]

Lipocalin 2 regulates mitochondrial phospholipidome remodeling, dynamics, and function in brown adipose tissue in male miceReviewers' comments:

Reviewer #1 (Remarks to the Author):

Lcn2 is a PA binding protein localized to the MAM compartment and essential for mitochondrial dynamics regarding fission and fusion. Absence of Lcn2 as shown previously by the group led to an impaired brown functionality in response to activation via a beta-adrenergic stimulus. Furthermore, absence of Lcn2 led to the remodeling of the mitochondrial lipidome of several CL species and especially the acyl chain composition of CL leading to an enhance mitochondrial damage which might impair functionality. In addition, the authors demonstrate that relevant genes of the Kennedy pathway are deregulated concomitant with an upregulation of PUFA containing TAGs. The work is conducted to a high standard and the experiments are well controlled. The problem I see is that limited insights that can be gained from this work due to a lack of mechanistic experiments.

The phenotype of the Lcn2 mice has been reported previously, and the characterization of the ko mice is in part overlapping with the current manuscript. While this is not per se a problem, I would remove all redundant figures and cite the original work here.

The data demonstrating the Lcn2 is a PA binding protein is very interesting but exclusively based on one experiment using a lipid strip system. This would need to be validated by alternate experimental approaches to allow for a firm conclusion.

The link between the function as a PA binding protein and the changes in the mitochondrial lipidome are completely unclear. The authors demonstrate at many point changes in gene regulation and protein levels of key enzymes in lipid synthesis pathways. How are the two points connected. Is it based on changes in PPAR signaling as reported in their JBC paper? If yes how is this I regulated and what is the mechanistic link.

Reviewer #2 (Remarks to the Author):

This study reports that Lcn2 is a PA-binding protein localized at MAM and that the absence of Lcn2 in brown adipocytes caused the alterations of mitochondrial morphology and mitochondrial phospholipidome in response to CL316,243 treatment. The results from the lipid binding assay and comprehensive lipidomics were worthwhile in obtaining insight into the role of Lcn2 in (phospho) lipid metabolism in brown adipocytes. In addition to these, the data including the cellular localization of Lcn2, mitochondrial morphology, and signaling pathway analyses seem suggestive of the involvement of Lcn2 in several cellular metabolism/signaling pathways in brown adipocytes; however, the provided experimental evidence did not sufficiently cover the mechanistic details of each, which lets their conclusions ambiguous. For example, although the authors claim that altered phospholipidome in Lcn2 deficient cells was linked to the impairment of mitochondrial dynamics/function, which results in the

activated inflammatory response, data to explain this link were not provided. To fully support their conclusions, detailed investigations as listed below would be required; alternatively, the authors better tone down their conclusions or focus on several aspects by elaborating on the mechanisms.

Major points:

1. Mitochondrial morphology seems contradictory between in vitro and in vivo Lcn2 KO models. In vitro Lcn2 KO cells had fragmented mitochondria as the authors mentioned; however, in vivo Lcn2 KO mitochondria appeared enlarged maintaining organized cristate as if healthy/improved mitochondria. This point may better be largely considered and carefully discussed given that all lipidome assessments were done with in vivo model. Related to this, TEM pictures of only 4C exposure mice were shown; how would normal temperature mice serve as a control?
2. Additional experiments would be needed to validate the idea of mitochondrial dysfunction due to Lcn2 deficiency leading to the activated inflammatory response. Although the authors provided mRNA expression data of mitochondrial proteins and fatty acid oxidation (supplemental figure 5), these should be assessed at protein levels. In addition, any mitochondrial functional assessment would be great to support this idea, which may include ROS, membrane potential, oxygen consumption measurement, etc.
3. It is vague how the conclusion of the Kennedy pathway disruption in Lcn2 KO was drawn. Although the mRNA expression of some enzymes was shown (Figure 8), those did not appropriately cover the majority of enzymes involved in the Kennedy pathway. Also, it does not sound that any lipid alterations in Lcn2-KO mice (summarized in supplemental figure 8) are directly tied to the Kennedy pathway disruption. It would rather be reasonable to discuss focusing on PA metabolism, such as the expression of enzymes involved in the upstream and downstream of PA, the transfer of PA at MAM, and so on.
4. The measurement of the PA species should be extended broader to discuss the acyl-chain profiles and the total amount. It might give insight into the flow of lipid metabolism related to the observed changes in lipidome since PA is intermediate.
5. MAM existence and ER/mitochondrial co-localization of Lcn2 were only validated in LPS-stimulated models (Figure 1). This would better be assessed with CL316,243 treatment in both WT and Lcn2 KO brown adipocytes to reliably elaborate the link between the Lcn2's MAM-localization and its involvement in phospholipid metabolism, given that all lipidomics were done with CL316,243 treatment models.
6. Lipidomics methodology: the measurement information including such as ionization, m/z values, and tandem mass conditions should be provided appropriately for each lipid class and/or species.

Minor points:

1. Data presentation:

- a. For western blot data, quantification and statistics of the band signals should be done. Also, the size of each protein should be indicated, ideally with a protein size marker(s).
- b. PE data shown in supplemental figure 7 lacked quantitative information. This may better be shown as the other lipid data.

2. Figure 3A: Body weight should be subjected to statistics as the other data were done.

3. It looks like labels were missing in the following parts: Figure 1C, Figure 3H.

4. Cite literature appropriately, for example; Lines 91-93 (Additionally, there exist...); lines 128-129 (One of the most...).

5. It was not provided how the sums of MUFA and PUFA-CLs were calculated (Figure 6 D-F).

Reviewer #3 (Remarks to the Author):

The manuscript by Su et al. describes a mechanism where Lcn2, a PA binding protein, mediates mitochondrial bioenergetics via regulating MAM phospholipid metabolism. In conditions of Lcn2 deficiency, mitochondrial fragmentation increases which leads to inflammation.

Major concerns:

Cardiolipin (CL) and phosphatidic acid (PA) comprise minority of

total mitochondrial phospholipids, does the Lcn2 deficiency induced remodeling of mitochondrial CL and PA not the main result of increased mitochondrial fragmentation? The reason why Lcn2 knockout mice responding to inflammation different from WT needs to be explained in Fig1, as CL316 243 injection model cannot mimic inflammation model perfectly.

Other concerns:

1. How about Lcn2 levels in other cell components as Lcn2 increased in whole cell lysates after LPS stimulation in Fig.1A?

2. In Fig2B, why the DRP1 phosphorylation at Ser616 much lower in *lcn2* deficiency condition than in WT as *lcn2* deficiency increased mitochondrial fragmentation? Is DRP1 phosphorylation at Ser616 a marker of increased mitochondrial fragmentation?

3. In Fig3E, the serum insulin level changed, did CL316,243 injection change islet function?

That's all for my questions. The phospholipidome part is clear and detailed.

Responses to Reviewers' Comments

Reviewers' Comments:

Reviewer #1 (Remarks to the Author):

Lcn2 is a PA binding protein localized to the MAM compartment and essential for mitochondrial dynamics regarding fission and fusion. Absence of Lcn2 as shown previously by the group led to an impaired brown functionality in response to activation via a beta-adrenergic stimulus. Furthermore, absence of Lcn2 led to the remodeling of the mitochondrial lipidome of several CL species and especially the acyl chain composition of CL leading to an enhance mitochondrial damage which might impair functionality. In addition, the authors demonstrate that relevant genes of the Kennedy pathway are deregulated concomitant with an upregulation of PUFA containing TAGs. The work is conducted to a high standard and the experiments are well controlled. The problem I see is that limited insights that can be gained from this work due to a lack of mechanistic experiments.

RESPONSES: Thanks the reviewer#1 for the positive comments on our manuscript **“The work is conducted to a high standard and the experiments are well controlled”**. We also appreciate the reviewer’s critical and constructive comments.

During the past six months, we have been working very hard to revise the manuscript. Per reviewers’ suggestions, we have conducted many new experiments, including *in vivo* and *in vitro* experiments, lipidomics, TEM, and ROS/Seahorse respiratory analysis. We added to the manuscript a substantial amount of new data and figures, including **Fig 1C, Fig 1H, Fig 2D, Fig 4H, Fig 5A-5O, Fig 6A-6H, Fig 6J-6M, Fig 7A-7S, Fig 8A, Fig 8C-8H, Supplemental Fig 5H-5J, Supplemental Fig 6A-6I, Supplemental Fig 7F, Supplemental Fig 7H-7J, and Supplemental Fig 9A-9Q, and Supplemental Fig 10**. We incorporated these new data into the manuscript, reorganized the Figures, and substantially revised all of the sections in the manuscript, particularly the results and discussion sections. With these new data, we were able to provide mechanistic insights into the role of Lcn2 as a PA binding protein in the LC-PUFA remodeling of phospholipids and mitochondrial function. Mechanistically, Lcn2 deficiency alters signaling PA activity through affecting PA production and function, which then deregulates the phospholipid-metabolizing enzymes that require PA for activation and function.

Based on our new findings, we additionally created three new diagrams to help with the insightful interpretation of the data. Below are the three new diagrams (Fig 8H, Supplemental Fig 6I, and Supplemental Fig 10: a new graphic summary) and a summary description that we added to the results and discussion sections.

“In summary, we have demonstrated that Lcn2 plays a critical role as a novel PA-binding protein in the homeostatic regulation of PA production and function, by which Lcn2 regulates the LC-PUFA remodeling of phospholipids, the peroxisomal LC-PUFA metabolism and plasmalogen biosynthesis, as well as mitochondrial dynamics and function. As indicated in a graphic summary (Supplemental Fig 10), Lcn2 deficiency alters signaling PA activity through affecting PA production and function, which then deregulates the phospholipid-metabolizing enzymes that require PA for activation and function. This disrupts the remodeling of mitochondrial phospholipid CL leading to an increase in the ratio of PUFA- to MUFA-containing CLs and the oxidation of CLs, ultimately contributing to mitochondrial damage and dysfunction. PA activation of Lcn2 deficiency also leads to mTOR signaling pathway activation, which additionally contributes to mitochondrial dysfunction and oxidative stress, leading to compensatory

activation of peroxisome function and increased LC-PUFA-containing plasmalogen biosynthesis. Changes in LC-PUFA levels also alters the production of lipid ligands required for the activation of transcriptional factors such as PPAR α , additionally contributing to mitochondrial dysfunction. Finally, dysfunctional mitochondria fail to maintain redox balance, leading to increased ROS levels and oxidative stress under inflammatory and metabolic stimulation. However, further investigations are warranted to explore the role of Lcn2 in mitochondria-peroxisome communication and whether PA serves as a signaling message connecting Lcn2-regulated functional compensation between these two organelles during metabolic stress."

Fig. 8H

Supplemental Fig 6I

Supplemental Fig 10

In terms of the novelty of this manuscript, we have identified Lcn2 as a novel PA binding protein and explored its role in mitochondrial phospholipid metabolism and PA signaling pathways. Indeed, this manuscript is the mechanistic extension (studies) of our previous published work. In our previous work, we showed that Lcn2 is a critical regulator of thermogenesis and mitochondrial function in BAT and beige adipose tissue. This manuscript mechanically uncovers that Lcn2 is located in mitochondria, MAMs, and acts as a novel PA binding protein to regulate mitochondrial phospholipid metabolism particularly the acyl-chain remodeling of cardiolipin a phospholipid unique to mitochondrial membranes. Lcn2 regulates PA production and function in controlling the activation of mTOR and ERK signaling pathways. Lcn2 deficiency alters PA production and function as a recursive regulator of PA-producing enzymes, deregulates PA signaling pathway activity, activates mitochondrial dysfunction-associated inflammation, and increases oxidative stress in BAT. This is the discovery of a novel mechanism for how Lcn2 functions in regulating mitochondrial function and thermogenesis. We believe this revised manuscript largely answers the questions regarding the mechanism for how Lcn2 works to regulate thermogenesis and mitochondrial function in adipose tissue as well as metabolic homeostasis and obesity/diabetes that others and we have reported previously.

The phenotype of the Lcn2 mice has been reported previously, and the characterization of the ko mice is in part overlapping with the current manuscript. While this is not per se a problem, I would remove all redundant figures and cite the original work here.

RESPONSES: The data we provided in this manuscript does not overlap with our previous publications. Although we did previously publish the TEM of mitochondrial morphology, in this manuscript the TEM is focusing on the morphology of mitochondrial swelling and the quantitative data of mitochondrial size that we have not previously reported. Indeed, we added the new data of TEM from BAT of mice at room temperature. Again, this data has not been published previously.

The data demonstrating the Lcn2 is a PA binding protein is very interesting but exclusively based on one experiment using a lipid strip system. This would need to be validated by alternate experimental approaches to allow for a firm conclusion.

RESPONSES: Thanks the reviewer#1 for pointing this out. We have conducted a new experiment using a different approach, i.e. protein pull down assay with PA-coated beads from Echelon Biosciences and added the new result as **Fig 1H** to validate the Lcn2-PA binding property. The new result has provided additional evidence supporting the binding of Lcn2 to PA.

The link between the function as a PA binding protein and the changes in the mitochondrial lipidome are completely unclear. The authors demonstrate at many point changes in gene

regulation and protein levels of key enzymes in lipid synthesis pathways. How are the two points connected. Is it based on changes in PPAR signaling as reported in their JBC paper? If yes how is this I regulated and what is the mechanistic link.

RESPONSES: Thanks! This is a very critical comment. As responded above, we have conducted several new experiments and provided a substantial amount of new data to help understand how Lcn2 functions as a PA binding protein in regulating the remodeling of mitochondrial phospholipids and mitochondrial function. Based on our new findings, we created three new diagrams to summarize and interpret our data (**Fig. 8H, Supplemental Fig 6I, and Supplemental Fig 10**). We believe that our new version of the manuscript with the incorporation of the new data largely answers the questions regarding the link of Lcn2 function as a PA binding protein to the remodeling of mitochondrial phospholipid cardiolipin and mitochondrial function in adipose tissue. For more details, please refer to the response to the first point.

As mentioned, PA serves as a precursor for the biosynthesis of cardiolipin and regulates the remodeling process of cardiolipin. Additionally, PA plays a role as a signaling molecule in the regulation of multiple pathways of lipid metabolism including TAG and phospholipids (PG and PE). Cardiolipin is a phospholipid unique to mitochondrial membranes and plays a critical role in mitochondrial membrane structure and curvature, thereby regulating mitochondrial function. For instance, Barth syndrome (BTHS) is a rare X-linked genetic disorder caused by mutations in the transacylase tafazzin gene encoding a mitochondrial enzyme that remodels the acyl chain composition of newly synthesized cardiolipin. Barth syndrome is characterized by changes in mitochondrial phospholipid structure and metabolism particularly the loss of cardiolipin, leading to mitochondrial dysfunction and severe cardiomyopathy. Thus, it is reasonable to connect a protein that interacts with PA to its role in mitochondrial function in adipose tissue.

Reviewer #2 (Remarks to the Author):

This study reports that Lcn2 is a PA-binding protein localized at MAM and that the absence of Lcn2 in brown adipocytes caused the alterations of mitochondrial morphology and mitochondrial phospholipidome in response to CL316,243 treatment. The results from the lipid binding assay and comprehensive lipidomics were worthwhile in obtaining insight into the role of Lcn2 in (phospho) lipid metabolism in brown adipocytes. In addition to these, the data including the cellular localization of Lcn2, mitochondrial morphology, and signaling pathway analyses seem suggestive of the involvement of Lcn2 in several cellular metabolism/signaling pathways in brown adipocytes; however, the provided experimental evidence did not sufficiently cover the mechanistic details of each, which lets their conclusions ambiguous. For example, although the authors claim that altered phospholipidome in Lcn2 deficient cells was linked to the impairment of mitochondrial dynamics/function, which results in the activated inflammatory response, data to explain this link were not provided. To fully support their conclusions, detailed investigations as listed below would be required; alternatively, the authors better tone down their conclusions or focus on several aspects by elaborating on the mechanisms.

RESPONSES: Thanks the reviewer#2 for the critical and constructive comments. Thanks the reviewer#2 for the positive comments "**The results from the lipid binding assay and comprehensive lipidomics were worthwhile in obtaining insight into the role of Lcn2 in (phospho) lipid metabolism in brown adipocytes**". We appreciate the reviewer#2's comments on "however, the provided experimental evidence did not sufficiently cover the mechanistic details of each, which lets their conclusions ambiguous."

We have been working very hard to revise the manuscript during the past six months. We have conducted many new experiments as suggested by the reviewers, including *in vivo* and *in vitro* experiments, new lipidomics analysis, TEM, and ROS/Seahorse respiratory analysis. We have obtained a substantial amount of new data and added the new data to the manuscript as new figures, including **Fig 1C, Fig 1H, Fig 2D, Fig 4H, Fig 5A-5O, Fig 6A-6H, Fig 6J-6M, Fig 7A-7S, Fig 8A, Fig 8C-8H, Supplemental Fig 5H-5J, Supplemental Fig 6A-6I, Supplemental Fig 7F, Supplemental Fig 7H-7J, and Supplemental Fig 9A-9Q, and Supplemental Fig 10.**

Accordingly, we substantially revised all of the sections in the manuscript, particularly the results and discussion sections. Based on our new findings, we additionally created three new diagrams to help with the insightful interpretation of the data. Please refer to the RESPONSE to the Reviewer#1's first point for the details. We believe that our new version of the manuscript with the incorporation of the new data largely answers the questions regarding the link of Lcn2 function as a PA binding protein to the remodeling of mitochondrial phospholipid cardiolipin and mitochondrial function.

Major points:

1. Mitochondrial morphology seems contradictory between *in vitro* and *in vivo* Lcn2 KO models. *In vitro* Lcn2 KO cells had fragmented mitochondria as the authors mentioned; however, *in vivo* Lcn2 KO mitochondria appeared enlarged maintaining organized cristate as if healthy/improved mitochondria. This point may better be largely considered and carefully discussed given that all lipidome assessments were done with *in vivo* model. Related to this, TEM pictures of only 4C exposure mice were shown; how would normal temperature mice serve as a control?

RESPONSES: Thanks the reviewer#2 for the positive comments "**The results from the lipid binding assay and comprehensive lipidomics were worthwhile in obtaining insight into the role of Lcn2 in (phospho) lipid metabolism in brown adipocytes**". Regarding the reviewer#2's question: "Mitochondrial morphology seems contradictory between *in vitro* and *in vivo* Lcn2 KO models", our answer is as follows. The *in vitro* and *in vivo* results are obtained using different methods. In brown adipocytes *in vitro*, cells were stained with Mitotracker red and observed under the confocal microscope, whereas the results of *in vivo* studies was from the TEM of electronic microscope. The morphology results from these two methods are not comparable. For example, under the confocal microscope, what we look at is the mitochondrial network (clusters). The tubular and punctuate are the different patterns of how mitochondria cluster to form a network (each cluster consists of many mitochondria). However, the TEM has much higher resolution, which allow us to see individual mitochondrion. Indeed, a study from the literature shows the similar phenomenon (see the figure extracted from the paper) as we saw regarding the mitochondrial swelling and fragmentation. They showed that ALCAT1 expression leads to mitochondrial fragmentation (Fig A and B) and enlargement (Fig F and G) (Li J et al. Proc Natl Acad Sci U S A. 2012).

Reference: Li J, Liu X, Wang H, Zhang W, Chan DC, Shi Y. Lysocardiolipin acyltransferase 1 (ALCAT1) controls mitochondrial DNA fidelity and biogenesis through modulation of MFN2 expression. Proc Natl Acad Sci U S A. 2012 May 1;109(18):6975-80. doi:10.1073/pnas.1120043109. Epub 2012 Apr 16. PMID: 22509026; PMCID: PMC3345005.

For the second part of comments, we performed a new TEM experiment to provide the control mitochondrial morphology as well as the quantitation of mitochondrial size at the room temperature in **Fig 2D**.

2. Additional experiments would be needed to validate the idea of mitochondrial dysfunction due to Lcn2 deficiency leading to the activated inflammatory response. Although the authors provided mRNA expression data of mitochondrial proteins and fatty acid oxidation (supplemental figure 5), these should be assessed at protein levels. In addition, any mitochondrial functional assessment would be great to support this idea, which may include ROS, membrane potential, oxygen consumption measurement, etc.

RESPONSES: Thanks! Per the reviewer#2's suggestion, we have conducted new experiments including Seahorse respiratory analysis, measurements of ROS and mitochondrial membrane

potential, as well as the assessments of anti-oxidant genes in differentiated brown adipocytes. The new results showed that Lcn2 deficient brown adipocytes have decreased OCR and maximal respiratory capacity, increased ROS and decreased MMP, as well as decreased anti-oxidant gene expression. We added the new data to the manuscript as a new Supplemental Fig 9 (see below).

3. It is vague how the conclusion of the Kennedy pathway disruption in Lcn2 KO was drawn. Although the mRNA expression of some enzymes was shown (Figure 8), those did not appropriately cover the majority of enzymes involved in the Kennedy pathway. Also, it does not sound that any lipid alterations in Lcn2-KO mice (summarized in supplemental figure 8) are directly tied to the Kennedy pathway disruption. It would rather be reasonable to discuss focusing on PA metabolism, such as the expression of enzymes involved in the upstream and downstream of PA, the transfer of PA at MAM, and so on.

RESPONSES: Thanks! This is a great point. After reanalyzing the lipidomics data in combination with new data, we made very interesting new discoveries. We replotted/reorganized the data/figures (Fig 4H, Fig 5A-5O, Fig 6A-6M), and created two new diagrams (Fig 8H and Supplemental Fig 6I see below) to clearly present/interpret the pathways that PA is involved. Additionally, we conducted new experiments with a new set of mice treated with saline or CL316, 243 for 14 days. Mitochondrial lipids were extracted from this new set of mice for the lipidomics analysis of PA and DAG levels. We also measured the mRNA expression of genes involved in the PC/PE synthesis via Kennedy and PEMT pathways as well as DAG-PA production. The new data was added to the manuscript as Fig 7A-7H and Supplemental 6A-6H. Moreover, we discovered that a group of ether phospholipids (plasmalogens) were significantly increased in Lcn2 KO BAT after CL316, 243 treatment, so did the protein levels of FAR1 a rate-limiting enzyme for plasmalogen biosynthesis. This indicates that mitochondrial dysfunction leads to compensatory activation of peroxisomes and increased PUFA levels. We added the new data as Fig 6A-6M (see below). Accordingly, we revised the relevant results section and discussion section substantially.

4. The measurement of the PA species should be extended broader to discuss the acyl-chain profiles and the total amount. It might give insight into the flow of lipid metabolism related to the observed changes in lipidome since PA is intermediate.

RESPONSES: Thanks! Per the reviewer#2's suggestion, we conducted a new experiment using a new set of mice treated with saline or CL316,243 for 14 days as described in the response to the point#3 above. Crude Mitochondria were isolated for lipid extraction and subsequent lipidomic analysis. We were able to detect a total of 6 PA species. Since DAG-PA production is the very important pathway that contributes significantly to PA production and abundance, we additionally measured DAG levels. We were able to detect 26 DAG species. Additionally, we determined the expression of DGK isoforms to indicate the changes in DAG-PA production. All the new data have been added to the manuscript as a new Fig 7. This new information helps us interpret the data better and understand deeper the role of Lcn2 in phospholipid metabolism.

5. MAM existence and ER/mitochondrial co-localization of Lcn2 were only validated in LPS-stimulated models (Figure 1). This would better be assessed with CL316,243 treatment in both WT and Lcn2 KO brown adipocytes to reliably elaborate the link between the Lcn2's MAM-localization and its involvement in phospholipid metabolism, given that all lipidomics were done with CL316,243 treatment models.

RESPONSES: We agree with the reviewer#2's comments. We have conducted the new experiment with a new set of mice treated with CL316, 243 for 14 days as well as 6 hours. MAM fractions isolated from pooled BAT from these mice (n=4) and Western-blotting was conducted. Our new results showed that Lcn2 protein was upregulated by CL316, 243 treatment at the MAM. The new data has been added to the manuscript as Fig 1C and Fig 6M.

6. Lipidomics methodology: the measurement information including such as ionization, m/z values, and tandem mass conditions should be provided appropriately for each lipid class and/or species.

RESPONSES: We have added the measurement information as the reviewer suggested and cited the reference as follows.

“Ionization voltages of -1.1, -0.95, and +1.2 kV and gas pressures of 0.3, 0.15, and 0.3 psi were employed on the nanomate apparatus for the analyses of anionic lipids, PE, and PC, respectively. The first and third quadrupoles were used as independent mass analyzers with a mass resolution setting of 0.7 Thomson, whereas the second quadrupole served as a collision

cell for tandem mass spectrometry (MS/MS) as previously described⁷⁸.”

Minor points:

1. Data presentation:

a. For western blot data, quantification and statistics of the band signals should be done. Also, the size of each protein should be indicated, ideally with a protein size marker(s).

RESPONSE: This has been done and the quantification data (only having significant difference) has been added to Fig 6, Fig 7, Fig 8, Supplemental Fig 5, and Supplemental Fig 7

b. PE data shown in supplemental figure 7 lacked quantitative information. This may better be shown as the other lipid data.

RESPONSE: The original Supplemental Fig 7 on PE data has been removed and replaced by new figures (Fig 5G and Fig 6D-6H) on PE information and quantitative information has been provided.

2. Figure 3A: Body weight should be subjected to statistics as the other data were done.

RESPONSE: This has been fixed.

3. It looks like labels were missing in the following parts: Figure 1C, Figure 3H.

RESPONSE: This has been fixed.

4. Cite literature appropriately, for example; Lines 91-93 (Additionally, there exist...); lines 128-129 (One of the most...).

RESPONSE: The appropriate references have been cited in these two places.

5. It was not provided how the sums of MUFA and PUFA-CLs were calculated (Figure 6 D-F).

RESPONSES: We have added the following information regarding how to calculate MUFA-CLs and PUFA-CLs to the figure legend of Fig 4.

“Sum of MUFA-CLs was calculated from the addition of the abundance of 11 measurable MUFA CLs (16:1-16:1-16:1-16:1, 16:0-16:1-16:1-16:1, 16:1-16:1-16:1-18:1, 16:0-16:1-16:1-18:1, 16:0-16:0-16:1-18:1, 18:1-18:1-16:1-16:1, 18:1-18:1-16:0-16:1, 18:1-18:1-16:0-16:0, 18:1-18:1-17:1-16:1, 16:1-18:1-18:1-18:1, 16:1-18:1-18:1-18:0). Sum of C18:2n6PUFA-CLs was calculated from the addition of the abundance of 7 measurable C18:2n6PUFA-CLs (18:2-18:2-16:1-16:1, 18:2-18:2-18:2-16:1, 18:2-18:2-18:2-16:0, 18:2-18:1-18:1-16:1, 18:2-18:2-18:2-18:2, 18:1-18:2-18:2-20:1, and 18:2-18:2-18:2-18:1). Sum of C20-22 PUFA-CLs was calculated from the addition of the abundance of 6 measurable C20-22n6 PUFA-CLs (20:4-18:2-16:1-16:1, 18:2-18:2-18:2-20:4, 18:2-18:2-18:2-20:3, 18:2-18:2-18:2-20:2, 18:1-18:2-18:2-20:2, and 18:2-18:2-18:2-22:5) and 2 measurable C22:6 PUFA CLs (18:2-18:2-16:1-22:6 and 18:2-18:2-18:2-22:6). The percentage of MUPA-CLs, C18-2n6 PUFA CLs, and C20-22n6 PUFA CLs was calculated from the sum of each category of CLs/total measurable CLs.”

Reviewer #3 (Remarks to the Author):

The manuscript by Su et al. describes a mechanism where Lcn2, a PA binding protein, mediates mitochondrial bioenergetics via regulating MAM phospholipid metabolism. In conditions of Lcn2 deficiency, mitochondrial fragmentation increases which leads to inflammation.

Major concerns:

Cardiolipin (CL) and phosphatidic acid (PA) comprise minority of total mitochondrial phospholipids, does the Lcn2 deficiency induced remodeling of mitochondrial CL and PA not the main result of increased mitochondrial fragmentation? The reason why Lcn2 knockout mice responding to inflammation different from WT needs to be explained in Fig1, as CL316 243 injection model cannot mimic inflammation model perfectly.

RESPONSES: We appreciate the Reviewer#3's critical comments. This question is similar to one of the reviewer#1's question and the Reviewer# 2's question #5.

As responded to the Reviewer#1 and #2 above, we have conducted many new experiments and provided a substantial amount of new data to help understand how Lcn2 functions as a PA binding protein in regulating the remodeling of mitochondrial phospholipids and mitochondrial function. Based on our new findings, we created three new diagrams to summarize and interpret our data (**Fig. 8H, Supplemental Fig 6I, and Supplemental Fig 10**). We believe that our new version of the manuscript with the incorporation of the new data largely answers the questions regarding the link of Lcn2 function as a PA binding protein to the remodeling of mitochondrial phospholipid cardiolipin and mitochondrial function in adipose tissue. For more details, please refer to the response to the the Reviewer#1's first point.

As mentioned, PA serves as a precursor for the biosynthesis of cardiolipin and regulates the remodeling process of cardiolipin. Additionally, PA plays a role as a signaling molecule in the regulation of multiple pathways of lipid metabolism including TAG and phospholipids (PG and PE). Cardiolipin is a phospholipid unique to mitochondrial membranes and plays a critical role in mitochondrial membrane structure and curvature, thereby regulating mitochondrial function. For instance, Barth syndrome (BTHS) is a rare X-linked genetic disorder caused by mutations in the transacylase tafazzin gene encoding a mitochondrial enzyme that remodels the acyl chain composition of newly synthesized cardiolipin. Barth syndrome is characterized by changes in mitochondrial phospholipid structure and metabolism particularly the loss of cardiolipin, leading to mitochondrial dysfunction and severe cardiomyopathy. Thus, it is reasonable to connect a protein that interacts with PA to its role in mitochondrial function in adipose tissue.

Our results have shown that Lcn2 deficiency increases the LC-PUFA but decreases MUFA remodeling of CLs leading to increased PUFA-CL to MUFA-CL ratio and mitochondrial damage. These changes are likely caused by alterations in PA production and function in regulating phospholipid-metabolizing enzymes (see Supplemental Fig 10 a graphic summary and RESPONSE to the Reviewer#1's first question).

We agree with the reviewer#3 that "CL316 243 injection model cannot mimic inflammation model perfectly". We have conduct the new experiment with a new set of mice treated with CL316,243 for 14 days as well as 6 hours. MAM fractions isolated from pooled BAT from these mice (n=4) and Western-blotting was conducted. Our new results showed that Lcn2 protein was upregulated by CL316, 243 treatment at the MAM. The new data has been added to the

manuscript as **Fig 1C and Fig 6M**. Please also refer to the RESPONSE to the Reviewer#2's question #5 for more information.

Other concerns:

1. How about Lcn2 levels in other cell components as Lcn2 increased in whole cell lysates after LPS stimulation in Fig.1A?

RESPONSE: Thanks! We conducted a new experiment looking at the Lcn2 protein levels in different cellular fractions (cytosolic, crude mitochondria, pure mitochondria, and MAM) and added the result as **Fig 1C**. Please refer to the RESPONSE to the Reviewer#2's question #5 for the results of Fig 1C.

2. In Fig2B, why the DRP1 phosphorylation at Ser616 much lower in Lcn2 deficiency condition than in WT as Lcn2 deficiency increased mitochondrial fragmentation? Is DRP1 phosphorylation at Ser616 a marker of increased mitochondrial fragmentation?

RESPONSES: Thanks! This is a great point. The balance between fusion and fission controls mitochondrial dynamics. Increased fragmentation of mitochondria can result from either increased fission or decreased fusion. For instance, a study reported that ALCAT1 expression can cause mitochondrial fragmentation and swelling by reducing MFN2 (fusion protein) (Li J et al. Proc Natl Acad Sci U S A. 2012). Thus, changes in Drp1 phosphorylation particularly detected with whole cell lysate may not necessarily be indicative of mitochondrial fragmentation or fission levels. Indeed, we have provided the data in **Supplemental Fig 7F** to show that Drp1 in MAM was slightly decreased in Lcn2 KO BAT after CL316, 243 treatment, but MFN2 in MAM was reduced in Lcn2 KO BAT. This supports that Lcn2 KO BAT has reduced mitochondrial fusion leading to increased fragmentation/fission.

3. In Fig3E, the serum insulin level changed, did CL316,243 injection change islet function? That's all for my questions. The phospholipidome part is clear and detailed.

RESPONSES: In the literature, acute CL316,243 injection did increase insulin levels and reduce blood glucose levels. This increase is accompanied by increased fatty acids. It is likely the fatty acids mediate CL316, 243-stimulated insulin secretion.

REVIEWERS' COMMENTS

Reviewer #1 (Remarks to the Author):

The revised manuscript is substantially improved most importantly through the addition of many new experiments. All of my concerns were addressed in the revision.

Reviewer #2 (Remarks to the Author):

The authors have appropriately addressed most of my original comments. I have only one minor concern related to the third major point that I originally raised. The interpretation of the PEMT pathway and the Lands' cycle (L334-337) may need to be reconsidered in light of the following points:

For the PEMT part, the authors should cite an appropriate reference that describes its relevance to PUFA-containing PC synthesis, such as PMID: 9370326.

LPCATs in the Lands' cycle have different substrate preferences. LPCAT1 prefers saturated acyl-CoA while LPCAT3 and LPCAT4 prefer PUFAs.

Reviewer #3 (Remarks to the Author):

I am pleased that the paper has been improved a lot. There are several questions need to be answered as follows:

1. In figure 1C, b-actin should be beta-actin.
2. In figure 1D, compared to pre-adipocyte and adipocyte, is there any difference in distribution of Lcn2 as in adipocyte Lcn2 looks more disperse.
3. As protein level of Lcn2 responses to inflammation like LPS, how about Lcn2 knockout or overexpressed cell lines or mice responses to inflammation?

REVIEWERS' COMMENTS

Reviewer #1 (Remarks to the Author):

The revised manuscript is substantially improved most importantly through the addition of many new experiments. All of my concerns were addressed in the revision.

RESPONSE: Thanks!!!

Reviewer #2 (Remarks to the Author):

The authors have appropriately addressed most of my original comments. I have only one minor concern related to the third major point that I originally raised. The interpretation of the PEMT pathway and the Lands' cycle (L334-337) may need to be reconsidered in light of the following points:

For the PEMT part, the authors should cite an appropriate reference that describes its relevance to PUFA-containing PC synthesis, such as PMID: 9370326.

LPCATs in the Lands' cycle have different substrate preferences. LPCAT1 prefers saturated acyl-CoA while LPCAT3 and LPCAT4 prefer PUFAs.

RESPONSE: According to the reviewer#2's comments, we cited the reference PMID: 9370326 as reference#39 and revised the text as follows.

“However, PEMT responsible for the synthesis of PUFA-containing PCs in the PEMT pathway as well as LPCATs **particularly LPCAT4** in the Land's cycle for the remodeling of C20:4-containing PC₃₈ had significantly higher....”

Reviewer #3 (Remarks to the Author):

I am pleased that the paper has been improved a lot. There are several questions need to be answered as follows:

1. In figure 1C, b-actin should be beta-actin.

RESPONSE: this has been fixed

2. In figure 1D, compared to pre-adipocyte and adipocyte, is there any difference in distribution of Lcn2 as in adipocyte Lcn2 looks more disperse.

RESPONSE: It would be different to make a comparison between preadipocytes and adipocytes. The Lcn2 fluorescence “looks more disperse” could be due to the interference of lipid droplets in adipocytes

3. As protein level of Lcn2 responses to inflammation like LPS, how about lcn2 knockout or overexpressed cell lines or mice responses to inflammation?

RESPONSE: We have published previously that Lcn2 deficiency leads to increased inflammation in adipose tissue.